



# The PRIMAP-hist national historical emissions time series

Johannes Gütschow[1], M. Louise Jeffery[1], Robert Gieseke[1], Ronja Gebel[1], David Stevens[1],
Mario Krapp[2], and Marcia Rocha[2]

[1]Potsdam Institute for Climate Impact Research, Telegraphenberg A 31, 14473 Potsdam, Germany
[2]Climate Analytics, Friedrichstraße 231, Haus B, 10969 Berlin, Germany

*Correspondence to:* J. Gütschow (johannes.guetschow(at)pik-potsdam.de)

**Abstract.** To assess the history of greenhouse gas emissions and individual countries' contributions to emissions and climate change, detailed historical data is needed. We combine several published datasets to create a comprehensive set of emission pathways of each country and Kyoto gas covering the years 1850 to 2014 for all UNFCCC member states as well as most non-UNFCCC territories. The sectoral resolution is that of the main IPCC 1996 categories. Additional subsectors are available

for time series of $CO_2$ from energy and industry. Country resolved data is combined from different sources and supplemented using growth rates from region resolved sources and numerical extrapolations to complete the dataset. Regional deforestation emissions are downscaled to country level using estimates of the deforested area obtained from potential vegetation and simulations of agricultural land. In this paper, we discuss the data sources and methods used and present the resulting dataset including its limitations and uncertainties. The dataset is available from http://doi.org/10.5880/PIK.2016.003 and can be viewed on the

website accompanying this paper (www.pik-potsdam.de/primap-live/primap-hist/).

## 1  Introduction

The question of responsibility for climate change and its impacts plays a significant role in the UNFCCC[1] negotiations for a global agreement to limit the global mean temperature increase and avoid dangerous climate change. It is interlinked with the discussion about equitable access to sustainable development which forms the basis of different frameworks to assess if

climate targets put forward by countries reflect a "fair share" in the collective burden to reshape the economy towards emissions neutrality. The Brazilian delegation to the UNFCCC has put forward a framework that assesses a country's contribution to climate change by calculating the fraction of the total warming generated by that country's historical greenhouse gas emissions. This approach is explained in Miguez and Filho (2000) and has been quantified in Höhne et al. (2010); Elzen et al. (2013); Matthews et al. (2014) amongst others. Other effort sharing proposals use cumulative per capita emissions as a metric and

thus also need a detailed record of historical emissions by individual countries (Winkler et al. (2011); Baer et al. (2008); Bode (2004)). In 2001 the MATCH[2] expert group was set up by the UNFCCC to generate historical emissions time series for this purpose. The dataset which resulted from this effort proved very useful in the negotiations and the scientific community (Höhne et al. (2010)). It was updated in Elzen et al. (2013) with EDGAR v4.2 data to cover all gases and emission until 2010. Here

---

[1]United Nations Framework Convention on Climate Change

[2]Ad hoc group for the modeling and assessment of contributions of climate change, www.match-info.net



we present a historical emissions dataset with a finer sectoral resolution, newly available input data, and new and improved methods for the combination of datasets.

We build our time series from a range of publicly available data sources (see Section 2) which are prioritized based on their completeness and reliability – an approach that has also been taken by the IPCC to compile the historical dataset for the 5th
Assessment Report (IPCC (2014), Annex.II.9, Historical data). For each time series (country, gas, sector resolved) the lower priority sources are used as growth rates to extend the higher priority sources. Where no country data is available we use regional growth rates, growth rates from other sectors, and numerical extrapolation to complete the time series.

For land use emissions we use the approach introduced in Matthews et al. (2014) and downscale a regional dataset using estimates of deforested areas derived from simulations of potential vegetation and agricultural land.

The PRIMAP-hist dataset covers the six Kyoto greenhouse gases and gas groups. Independent time series are generated for carbon dioxide ($CO_2$), methane ($CH_4$), nitrous oxide ($N_2O$), hydrofluorocarbons (HFCs), perflurocarbons (PFCs) and sulfur hexafluoride ($SF_6$). For all gases except $CO_2$ the sectoral resolution is that of the main IPCC 1996 categories. For $CO_2$ more detailed categories are used because some important datasets cover only subsectors of categories 1 and 2. For details and sector names we refer to Table 1 9. $NF_3$ is not included as it has only been included into the group of Kyoto Protocol relevant gases
for the second commitment period of the Kyoto Protocol which started in 2013 and data availability is therefore still scarce.

The emissions time series cover the period of 1850 to 2014. This is achieved through the combination of various sources and extrapolation for some sectors, gases, and countries both into the past and into the future. The extent of the extrapolation needed varies between sectors, gases and countries. Data coverage is very good for energy related $CO_2$ emissions for the whole period. For other gases and sectors we have to rely on growth rates from regional data for the period before 1970. The
data sources we use are described in Section 2, while the details of the combination process, including the prioritization, are described in Section 4.

As this dataset is designed to be used for global climate policy analysis, we provide data for all 196 member states of the UNFCCC as well as several countries and territories that are not UNFCCC members, not internationally recognized, or associated with a UNFCCC member state but not included in the emissions reporting of that state. We follow the territorial
coverage of the countries' submissions to the UNFCCC and use territorial accounting which is in line with UNFCCC standards. Territorial accounting attributes emissions originating from a certain territory at any point in time to the state the territory currently belongs to. Emissions of former colonies are thus attributed to the now independent state and not to the former metropolitan state. Occupation of countries' territories is only taken into account if the occupying country reports the emissions from that territory.[3] In the supplementary information we present a list of territories included in the emissions of UNFCCC
Parties as well as information on the territories that are treated separately (Section C).

The paper is organized as follows: we begin by describing the individual data sources we use in Section 2 and their prioritization in Section 3. In Section 4 we describe how the dataset is constructed from the individual sources including the special treatment of land use data. In Section 5 we give information on how to obtain and use the data. Results are described in

---

[3]This is e.g. the case for Israel and the Palestinian Territory



Section 6 with information on the uncertainties of emissions data in Section 7. Limitations are covered in Section 8. Methodological details, sector coverage, territorial definitions and data sources which we did not use are described in the Appendix.

## 2   Data sources

In this section we describe the data sources used to create our composite source. We only use sources which are publicly available and prefer sources that are not composites of other sources to avoid including original sources twice, once directly and once indirectly through a composite source. However, it is likely that some sources share at least some input data such as information on fossil fuel production or use the same emission factors. The sources are grouped into four categories. **Country reported data** is the highest priority category as it can benefit from detailed knowledge about the specific situation in a country and is well accepted in the context of the UNFCCC negotiations. This is exemplified by the linking of the entry into force of the Paris agreement to the latest country reported emissions and not to any third party source (UNFCCC (2015b)). Where this data is not available or does not meet our minimum requirements (see Section 2.1 below) we use **country resolved data** provided by third parties like research institutions and international organizations. To extrapolate data into the past we use **region resolved datasets**. Finally, we use some **gridded datasets** and calculate country resolved data using country masks. Figure 1 gives an overview of the data sources described in detail in the remainder of this section. Detailed information on data preprocessing is available in Section D.

In the text we refer to data sources using the acronyms introduced in the source description below.

### 2.1   Minimum requirements for data

To be useful for our composite source, data has to meet some minimal requirements. Emissions data has inherent fluctuations due to weather (determining heating requirements), economic activity, and other factors. Not all sources model all these factors equally and therefore exhibit different fluctuations. When combining the sources, we use the growth rates from the lower priority source to extend a higher priority source. To weaken the influence of these fluctuations we use the trend of several years for the matching instead of a single year. We therefore require that each time series contains at least three data points spread over a period of at least 11 years. Furthermore, we need time series with the detail of sectors and gases listed in Table 9.

### 2.2   Country reported data

Under the UNFCCC there are several requirements for reporting of greenhouse gas emissions data (Yamin and Depledge (2005)). Under the convention both developed (Annex I) and developing (non-Annex I) Parties[4] have to regularly submit communications that include an inventory of national GHG emissions and removals. Detailed requirements, however, differ strongly between Annex I and non-Annex I Parties. Annex I Parties have to submit an inventory which covers all sectors, gases,

---

[4]The term Parties refers to the countries which have ratified the UNFCCC. Annex I Parties refers to those countries listed in Annex I of the Kyoto Protocol (KP) which are the developed countries under the UNFCCC. The definition is now almost two decades old and does not represent the state of economic development any more. The distinction between developed and developing countries is thus subject to constant debate in the UNFCCC meetings.




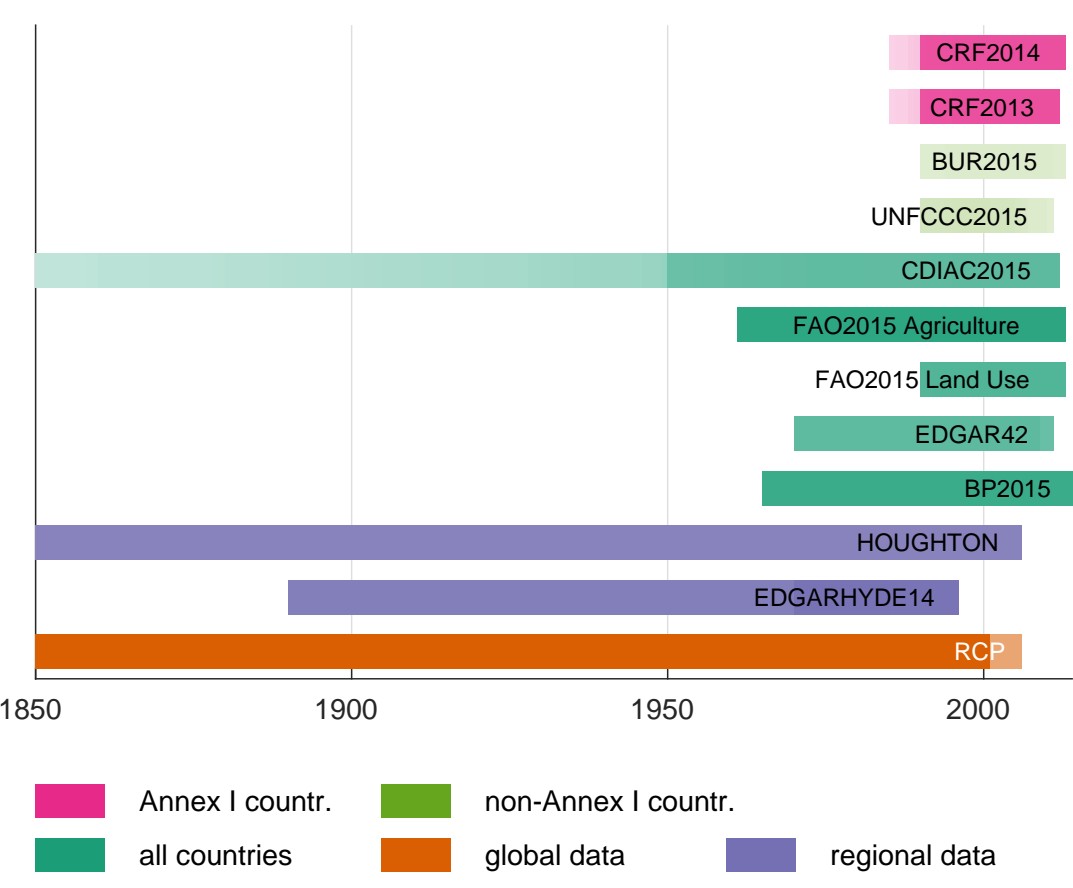

**Figure 1.** Coverage of years and countries in the sources used for the PRIMAP-hist dataset. The color indicates the country group covered or the regional resolution, while the intensities indicates the fraction of countries in the group covered by the source in each year. The fraction is taken over all gases and categories which can be seen in the CDIAC time series where the flaring time series only starts in 1950. RCP time series for $CH_4$ end in 2000 leading to the lower coverage after year 2000.





and years since 1990 annually. The submissions should consist of two parts, the common reporting format (CRF) tables with the data and a national inventory report (NIR) which gives background information like the rationale behind the selection of emission factors and methodological questions. For details on the CRF tables see Section 2.2.3 below. Annex I Parties also submit national communications which originally served the purpose to report on policies and measures to implement the

Party's commitment to aim to return emissions to 1990 levels by the year 2000. The NIRs have recently (decided in 2011 at COP17[5], Durban) been complemented with Biennial Reports to enhance reporting. The emissions data contained should be consistent with the CRF data. Under the KP Annex I Parties have to regularly submit information needed to assess whether they are meeting their emission targets. For our purpose the CRF data is the most useful of these sources. The other sources do not provide additional information for the purpose of this paper and are not used.

Non-Annex I Parties were required to submit an initial national communication within three years after the entry into force of the convention. The least developed countries (LDCs) could decide if they submitted an initial national communication. The submissions were required to contain an emissions inventory, which covers the years 1990 to 1994 for most submissions. A time frame for subsequent national communications could not be agreed upon, and only few countries submitted further national communications with updated inventories. The guidelines for national communications for non-annex I Parties are

less stringent than the guidelines for Annex I Parties, consequently the coverage and detail in sectors and gases of the data differs strongly between countries. Since 2014 non-Annex I Parties are required to report GHG inventory information through Biennial Update Reports (BUR). The first report was due by December 2014, however only 24 of over 150 countries actually submitted (as of January 2016). LDCs and SIDS[6] are exempted from the mandatory submission and can submit at their discretion.

The Paris agreement requires regular national inventory reports by all Parties which might improve emissions reporting in the future (UNFCCC (2015b), Article 7 (a)).

### 2.2.1   National Communications and National Inventory Reports for developing countries [UNFCCC2015]

Most developing countries reported historical emissions data at least once using National Communications (UNFCCC (2015a)) and sometimes National Inventory Reports. However, several countries only reported data for the period of 1990 to 1994,

sometimes only single years. Therefore, a lot of countries' submissions do not meet our minimal data requirements and are consequently not used for the composite source. Where the data meets our requirements we use it with high priority as it is prepared by in-country experts which gives the results based on it high credibility within the country, which is beneficial for policy analysis. We compare it with third party data to identify if there are differences that can not be explained by uncertainties. National Inventory Reports give a more detailed overview over the emissions inventory than national communications, but are

not published by all countries. While developed country Parties also submit National Communications and National Inventory Reports we only use this data for developing countries as we have the CRF data for developed countries (see Section 2.2.3 below). The data used here has been downloaded from the UNFCCC website using the "Detailed data by party" interface

---

[5]COP: Conference of the Parties to the UNFCCC
[6]Small Island Developing States




UNFCCC (2015c). The date of access was September 25th, 2015. Some countries submit their data prepared according to IPCC 2006 guidelines. This data is not available through the interface (Andorra, Cook Islands, Jamaica, Kiribati, Malawi, Mauritania, Mexico, Namibia, Samoa, Swaziland, South Africa and Tunisia). Furthermore, the detailed by party interface seems to lag behind the submissions and misses some submissions from 2015 and 2016 (as of February 1st 2016).

### 2.2.2 Biennial Update Reports [BUR2015]

Biennial Update Reports (BURs) are submitted to the UNFCCC by non-Annex I Parties (UNFCCC (2016)). They contain greenhouse gas emissions information with varying detail in sectors, gases and years. So far 24 countries have submitted data. Unfortunately most of the submissions do not meet our minimal data requirements.

Argentina, Ghana, India, Namibia, Paraguay, Peru, Thailand, Tunisia, and Vietnam submitted detailed values only for a single year. Bosnia published data for 2010 and 2011. Andorra and Macedonia published only aggregate Kyoto greenhouse gas data.

Brazil and Singapore published detailed information for 1994, 2000, and 2010, however, the level of detail is not sufficient for all sectors. Chile, Mexico, South Africa, South Korea, and Uruguay have detailed information for a range of years in the annex to the BUR and the NIR. However, for South Africa the level of detail is not sufficient for all sectors and gases.

Colombia, Costa Rica, and Montenegro use the IPCC 2006 categorization so we can not include the data in the current version of this dataset. The Lebanon BUR was not accessible on the UNFCCC website so we could not assess whether there is useful data in it (as of February 1st 2016).

The final PRIMAP-hist dataset uses BUR2015 data for Azerbaijan, Brazil, Chile, Republic of Korea, Mexico, Singapore, South Africa, and Uruguay.

### 2.2.3 UNFCCC CRF [CRF2014, CRF2013]

CRF data, short for Common Reporting Format, is reported by all Annex I Parties every year on a mandatory basis. The data is very detailed both in sectors and gases and undergoes review for consistency and compliance with reporting guidelines by expert teams from the UNFCCC roster of experts. We use the final version of the 2014 data (UNFCCC (2014a)) which contains information until the year 2012. The 2013 revision (UNFCCC (2013)) is used as a backup in case there are gaps in the 2014 data. The first year is 1990 with a few exceptions of data series starting in 1985. All Kyoto gases are covered, data is submitted using IPCC 1996 categories.[7]

The 2015 edition of the CRF data uses IPCC 2006 categories. This posed problems in data preparation for several countries such that publication was significantly delayed. To date (April 2016) still not all countries have submitted their data with large emitters missing.[8] The gas $NF_3$ has been added as it is included in the Kyoto greenhouse gases for the second commitment

---

[7]When we write "all" there can still be a few exceptions where data is missing for single countries or sectors.

[8]No 2015 CRF data has been submitted by Belarus and Switzerland. Canada and the United States have submitted data but requested to not make it publicly available until problems with the CRF reporter software are solved.



period of the Kyoto Protocol. CRF2015 data will be included in a future update of this dataset together with a move to IPCC 2006 categorization.

## 2.3 Country resolved data

### 2.3.1 BP Statistical Review of World Energy [BP2015]

The BP Statistical Review of world Energy is published every year and contains time series for $CO_2$ emissions from consumption of oil, gas, and coal. Emission data are derived on the basis of the carbon content of the fuels and statistics of fuel consumption. The 2015 edition (British Petroleum (2015)) contains information for 76 individual countries and 5 regional groups of smaller countries which we downscale to country level. The first year in the time series is 1965, the last is 2014.

### 2.3.2 CDIAC fossil $CO_2$ [CDIAC2015]

The CDIAC fossil fuel and industrial $CO_2$ emissions dataset is published by the Carbon Dioxide Information Analysis Center (CDIAC) with regular updates (Boden et al. (2015)). It covers emissions from fossil fuel burning, flaring, and cement production for 221 countries and territories. The first year is 1751 and the last year 2011. Emissions from 1751 to 1949 are computed using statistics of fossil fuel production and trade combined with information on the chemical composition and assumptions on the use and combustion efficiency following the methodology presented in Andres et al. (1999). Emission data for the years 1950 to 2011 are based primarily on the United Nations energy statistics using the methodology presented in Marland and Rotty (1984).

### 2.3.3 EDGAR versions 4.2 and 4.2 FT2010 [EDGAR42]

The EDGAR (Emissions Database for Global Atmospheric Research) dataset is published by the European Commission Joint Research Center (JRC) and Netherlands Environmental Assessment Agency (PBL). It undergoes regular updates. The current version is 4.2. It contains emissions data for all Kyoto greenhouse gases as well as other substances. It covers 233 countries and territories in all parts of the world, though not all countries have full data coverage. EDGAR version 4.2 (JRC and PBL (2011)) covers the period 1970 to 2008. Additionally the EDGAR v4.2 FT2010 (JRC and PBL (2013); International Energy Agency (2012)) covers the period 2000 to 2010 and EDGAR v4.2 FT2012 (JRC and PBL (2014); Unep (2014)) covers 1970 to 2012 but only for $CO_2$, $CH_4$, $N_2O$, and aggregate KyotoGHG emissions with no sectoral resolution. Version 4.3 covering the period until 2012 has been implicitly announced but is not yet available (as of February 1st 2016).[9]

EDGAR time series are calculated using activity data on a per sector, per gas and per country basis. Emissions are calculated using a country, sector, and gas specific technology mix with technology dependent emission factors. The emission factors for each technology are determined by end of pipe measurements, country specific factors, and a relative emission reduction factor to incorporate installed emissions reduction technologies.

---

[9]In the "Trends in Global $CO_2$ emissions report" (Olivier et al. (2015)) EDGAR v4.3 is referenced as forthcoming in 2015.



### 2.3.4   FAOSTAT database [FAO2015]

The Food and Agriculture Organization of the United Nations (FAO) publishes data on emissions from agriculture and land use (Food and of the United Nations (2015a)). Over 200 countries and territories are included in the database.

The land use emissions are categorized into forestland, grassland, cropland, and biomass burning where the first three categories contain information on $CO_2$ only, while biomass burning also contains information on $N_2O$ and $CH_4$ emissions. To generate the time series, data from land use and forestry databases (both from FAO and other institutions) are used together with IPCC estimates of emission factors and the FAO "Global Forest Resources Assessment" database for carbon stock in forest biomass. For details we refer to the methodology information on the FAOSTAT website (Food and of the United Nations (2015b)). The time series cover 1990 to 2012.

FAOSTAT data for agricultural emissions ranges from 1961 to 2012. It covers $N_2O$ and $CH_4$ from various sources. Because it covers a longer time period than other sources for the agricultural sector we use it with highest priority after the country reported data. The data is generated from activity data and emission factors following the Tier 1 approach of the IPCC 2006 guidelines.

FAO data does not follow the IPCC sector definitions, so sectors have to be mapped to the appropriate IPCC sectors.

## 2.4   Region resolved datasets

### 2.4.1   Houghton land use $CO_2$ [HOUGHTON2008]

This source covers land use $CO_2$ emissions from 7 regions and three individual countries (USA, Canada, and China). The dataset is described in a series of papers by Houghton (2008, 2003, 1999). It is generated using a book keeping model to track carbon in living vegetation, dead plant material, wood products, and soils. The carbon stock and its changes are taken from field studies. Information on changes in land use are mostly taken from agricultural and forestry statistics, historical records, and national handbooks. Emissions outside tropical regions past 1990 are estimates (constants). For our dataset the regional emissions have to be downscaled to country level.

### 2.4.2   RCP historical data [RCP]

The Representative Concentration Pathways (RCPs) were created for the CMIP5 intercomparison study of Earth System Models for the IPCC's Fifth Assessment Report (AR5). They have a common historical emission time series which covers all Kyoto gases but is only resolved at a coarse regional and sectoral level (Meinshausen et al. (2011)). RCP historical data are compiled from a wide range of emission sources and atmospheric concentration measurements. Where concentration data is used, inverse emission estimates are computed using the MAGICC6 reduced complexity climate model. RCP historical data can be used for extrapolation of country time series to the past using regional growth rates.



## 2.5 EDGAR-HYDE 1.4 [EDGAR-HYDE14]

The EDGAR-HYDE 1.4 "Adjusted Regional Historical Emissions 1890 – 1990" dataset covers the gases $CO_2$, $N_2O$, and $CH_4$ for the years 1890 to 1995 (Olivier and Berdowski (2001); Van Aardenne et al. (2001)). The data is given for 13 regions, some of which are individual countries (USA, Canada, Japan). It is generated from the EDGAR v3.2 dataset (Olivier and Berdowski

(2001)) and the "Hundred Year Database for Integrated Environmental Assessments" (HYDE) (Van Aardenne et al. (2001); Goldewijk and Battjes (1997)). We use it to extrapolate country emissions into the past. It has a relatively high sectoral detail, but the sectors differ from the IPCC 1996 definitions, so mapping to IPCC 1996 sectors is necessary.

## 2.6 Gridded datasets

### 2.6.1 HYDE land cover data [HYDE]

The HYDE land cover data (Klein Goldewijk et al. (2011, 2010); PBL (2015)) is generated using hindcast techniques and estimates on population development over the last 12,000 years. For the time period of interest here it provides estimates of pasture and crop land on a 5' resolution grid for 10 year time steps. It does not directly provide estimates for deforestation, but these can be computed by comparison with simulation data of potential vegetation.

### 2.6.2 SAGE Global Potential Vegetation Dataset [SAGE]

This dataset is available in the SAGE (Center for Sustainability and the Global Environment) database and is described in Ramankutty and Foley (1999) and available for download from Ramankutty and Foley (2015). It contains 5' resolution grid maps of potential vegetation (i.e. vegetation that potentially could be in a certain spot if there was no human interference) for a time period from 1700 to 1992. It has been used together with HYDE 3.1 in Matthews et al. (2014) to downscale CDIAC land use $CO_2$ emissions to country level. We use it for the same purpose here.

## 20  3   Source prioritization

To create a dataset covering all countries and gases for a period of over 150 years, multiple data sources need to be combined as no single source contains all the necessary data. We order sources such that the highest quality sources are selected for each gas, category, and year according to availability. Where possible, source prioritization is defined, and used, at a global level. The details of the process of the combination of sources are described in Section 4.

## 25  3.1   Emissions from energy, industrial processes, solvent use, agriculture, and waste

For fossil emissions our highest priority source is the UNFCCC CRF data as it is both accepted by the countries that report and by other countries as it is peer reviewed. However, it is only available for developed country Parties. We use CRF2014 and fill gaps with CRF2013 where necessary. For non-Annex I Parties we use data from National Communications and National Inventory Reports with highest priority (UNFCCC2015). For a few developing countries, data from the Biennial Update Reports

(BUR2015) is available and fulfills our minimal requirements. It is used to supplement The UNFCCC2015 data. UNFCCC2015 is prioritized over BUR2015 because the latter only contains a few data points for most countries while the UNFCCC2015 data contains full time series for more countries. Those sources of UNFCCC reported data cover a wide range of gases and sectors (for most countries $CO_2$, $CH_4$, and $N_2O$ for all sectors at the level of detail needed for the composite source. Fluorinated

gases are only contained for a few countries). For fossil fuel burning related $CO_2$, $CO_2$ from flaring, and $CO_2$ emissions from mineral products we use CDIAC as the next source. For $CO_2$ from other sectors and all other gases we use a combination of EDGAR v4.2 FT2010 and EDGAR v4.2 as the next source. It is also used to complement CDIAC data where necessary (e.g. for small countries missing in the CDIAC source). BP2015 data is used to extend the energy $CO_2$ time series until 2014. Where no country reported data is available the country resolved data sources are used as the first sources.

Sources without country level information, RCP and EDGAR-HYDE, are used to extrapolate emissions into the past. As EDGAR-HYDE has a higher regional and sectoral resolution it is used as the first priority source for extrapolation of $CO_2$, $CH_4$, and $N_2O$ emissions. Emissions from fluorinated gases for years before 1970 are only available from the RCP historical data and only on a global level.

The source prioritization for the individual gases is summarized in Tables 1, 2, 3, and 4. Details of the source creation

methods are available in Section 4.1.

### 3.2   Land use, land use change, and forestry emissions

The first priority sources are country reported data which are supplemented with FAOSTAT data. None of these sources contain information for the period before 1990. EDGAR42 does contain information starting in 1970 but excludes sinks from the calculation of $CO_2$ land use emissions which is why we exclude EDGAR $CO_2$ land use data from our dataset. The period before

1990 is covered by the Houghton dataset on a regional level, which we downscale using estimates of historical deforestation (see Section 4.2).

For $CH_4$, and $N_2O$ we use country reported data, FAOSTAT, and EDGAR data on a per country basis. Regional growth rates from EDGAR-HYDE14 are used to extrapolate the time series.

Source prioritization is summarized in Tables 5 and 6. Details of the source creation are presented in Section 4.2.

### 4   Dataset construction

### 4.1   Emissions from energy, industrial processes, solvent use, agriculture, and waste

The generation of the emissions time series is carried out using the Composite Source Generator (CSG) of the PRIMAP emissions module described in Nabel et al. (2011). Data is aggregated on a per country, per gas, and per category level taking into account source prioritization (see Section 3). The result is one time series for each country, category, and gas. The source

creation is organized in four steps described below.





**Source preprocessing** If data is defined on a more detailed level of gases (in case of HFCs and PFCs) or categories (e.g. categories 4A and 4B) it is aggregated to the resolution described above for all sources individually. The aggregate time series covers the union of all years of the individual gas or sector series. If data is missing for some years in some of the individual gases or subcategories it is interpolated to close gaps and extrapolated to fill missing data at the boundaries before aggregation. After aggregation, the information that a subcategory or gas was missing is lost. If data is missing on the gas and category level we are working with in the PRIMAP-hist dataset, it is not interpolated in preprocessing as it can be filled from other sources.

**Composite Source Generator** The composite source generator (CSG) works on every country, gas, and category individually. Its core is the priority algorithm which combines the sources following a given prioritization. The algorithm starts with the highest priority source. Missing time series are copied from lower priority sources. After this step the priority algorithm fills gaps in the time series using lower priority sources and extrapolates using growth rates from lower priority sources. For each gas, category, country time series it is checked if the composite source contains gaps or does not cover the full time period. If that is the case the second highest priority source is checked for data that could fill gaps and extend the time series. If that time series itself contains gaps or needs extension, the hierarchy is parsed downwards recursively and the resulting time series is used to extend the composite source. For details on the harmonization see Appendix A4. For this study we add one source at a time and therefore do not parse the sources recursively but add what is present in the next source and then see if the resulting time series needs further extension. If there is data missing after the end of this process the CSG can numerically interpolate gaps and extrapolate missing data at the boundaries. For this dataset we only use the interpolation by the CSG, because we use regional growth rates from other sources to extrapolate the country data. A schematic of the composite source generator within the PRIMAP emissions module is shown in Figure 2.

**Extrapolation** Missing years in the past are extrapolated using growth rates from regional data or data from other sectors and numerical extrapolation. The details depend on gas and sector and are described later in this section. Missing data in the future is extrapolated linearly using a 15 years trend. This usually affects up to 4 years with very few exceptions where extrapolation is used for longer periods. We also offer a dataset without this numerical extrapolation.

**Postprocessing** After extrapolation the individual gas and category time series are aggregated to build the higher categories and the Kyoto GHG basket. For details on the aggregation see Appendix A1.2.

Sudan needs a special treatment as the split into Sudan and South Sudan has been so recent that no separate emissions data is available yet. We downscale the Sudan emissions time series to Sudan and South Sudan using UN population statistics (UN Population Division (2015)) as a downscaling key. We also aggregate country data for some regional groups.

Figure 3 shows an example of how we build a pathway from different time series.

In the following we describe the data availability and use in detail for the different gases and sectors.



**PRIMAPDB**

EDGAR42

CDIAC2015

RCP

FAO2015

CRF2014

UN2012

Composite Source

...

**Preprocess**
* Translate to common sector terminology
* Downscale if neccessary
* Aggregate sectors and gases

**Prioritize sources**
Select and prioritize sources

**Start compsource**
Start composite source with a copy of the highest priotity source

**Copy missing countries**
Copy missing countries / regions to composite source if available in lower priority sources

**Calculate or copy missing categories**
If categories are missing
(1) aggregate sub-categories
or (2) copy, if available in lower priority sources

**Inter- and extrapolate over time (priority alg.)**
Complete the composite source time series by using growth rates from lower priority data

**Extrapolation**
Extrapolate time series using regional data or numerical methods.

**Figure 2.** The Composite source Generator is used to assemble time series from different sources into one time series covering all countries, sectors, gases and years. The source prioritization in the figure is illustrative and does not represent the source prioritization for the dataset described here. In this study the internal category aggregation of the composite source generator is not used but categories are aggregated before the generation of the composite source to enable extrapolation of subcategories.







**Figure 3.** Example for the work of the composite source generator: the creation of the category 1A, $CO_2$ pathway for Korea. The buildup starts with the UNFCCC source as there is no CRF data for Korea. Extrapolation is not needed in this case, so the step is omitted from the figure.



| Step | Source | Categories | Countries | Years | Type of operation |
|------|--------|-----------|-----------|-------|-------------------|
| 1 | CRF2014 | all | Annex I | 1990 - 2012 | CSG |
| 2 | CRF2013 | all | Annex I | 1990 - 2011 | CSG |
| 3 | UNFCCC2015 | all | 35 non-Annex I | 1990 - 2010 | CSG |
| 4 | BUR2015 | all | 8 non-Annex I | 1990 - 2012 | CSG |
| 5 | CDIAC2015 | 1A, 1B2, 2A | almost all | 1850 - 2011 | CSG |
| 6 | EDGAR42 | all | almost all | 1970 - 2010 | CSG |
| 7 | BP2015 | 1A | almost all | 1965 - 2014 | CSG |
| 8 | EDGAR-HYDE14 | 1A, 1B1-2, 2A-G | regions | 1890 - 1995 | growth rates extrap. |
| 9 | RCP | 1A | global | 1850 - 2005 | growth rates extrap. |
| 10 | PRIMAP-hist CAT1A | all but 1A | all | 1850 - 2014 | growth rates extrap. |
| 11 | numerical | all | all | 1850 - 2014 | linear extrapolation |

**Table 1.** Source prioritization and extrapolation for fossil and industrial $CO_2$. Years are maximal values. Some countries have less coverage. In CRF a few countries have data starting a few years before 1990.

**$CO_2$** Data coverage for $CO_2$ is in general very good. The largest emission sources are the consumption and production of fossil fuels and the production of cement. Both are covered by CDIAC which extends the country reported data back to 1850 for 31 countries, back to 1900 for 65 countries, back to 1950 for 168 countries and to 1990 for 196 countries. For other sectors EDGAR42 extends the time series back to 1970. BP data completes the fossil fuel consumption time series until 2014.

To further extend time series into the past we use EDGAR-HYDE regional growth rates (starting in 1890). For categories 1A, 1B1, and 1B2 explicit time series are available while we use category 2 time series as a proxy for the subcategories of category 2. Other categories are not available. RCP $CO_2$ data that ranges back until 1850 is only available for total emissions excluding LULUCF on a global level. As total $CO_2$ emissions are dominated by fossil fuel burning we use the RCP data as growth rates to extrapolate category 1A emissions for those countries which were not covered by CDIAC and EDGAR-HYDE from 1850 onwards. This does not affect any mayor emitter at the time for which data is extrapolated. For categories 3, 4, 6, and 7 no source for extrapolation is available so the first year is 1970 from EDGAR. We use growth rates of the the fossil fuel consumption time series for each country as a proxy to extend the time series of all other sectors to 1850.

The source prioritization and extrapolation is summarized in Table 1. Details of the growth rate extrapolation are discussed in Appendix A5.1.

**$CH_4$** We have data on a per country level from 1990 to 2010 or 2012 from the country reported data. For agriculture (category 4) we have FAOSTAT data where the first year is 1961 and the last year 2012. For all other sectors and missing countries we use EDGAR42 which covers 1970 to 2010 for almost all countries. Categories 1, 2, 4, and 6 are extrapolated to 1890



| Step | Source | Categories | Countries | Years | Type of operation |
|------|--------|-----------|-----------|-------|-------------------|
| 1 | CRF2014 | all | Annex I | 1990 - 2012 | CSG |
| 2 | CRF2013 | all | Annex I | 1990 - 2011 | CSG |
| 3 | UNFCCC2015 | all | 35 non-Annex I | 1990 - 2010 | CSG |
| 4 | BUR2015 | all | 7 non-Annex I | 1990 - 2012 | CSG |
| 5 | FAO2015 | 4 | almost all | 1961 - 2012 | CSG |
| 6 | EDGAR42 | all | almost all | 1970 - 2010 | CSG |
| 7 | EDGAR-HYDE14 | 1, 2, 4, 6 | regions | 1890 - 1995 | growth rates extrap. |
| 8 | RCP | 1, 2, 4 ,6 | global | 1850 - 2000 | growth rates extrap. |
| 9 | numerical | 7 | all | 1850 - 2010 | linear to zero in 1850 |
| 10 | numerical | all | all | 1850 - 2014 | linear extrapolation |

**Table 2.** Source prioritization for fossil and industrial $CH_4$. Years are maximal values. Some countries have less coverage. In CRF a few countries have data starting a few years before 1990. Note that there are no $CH_4$ emissions data in category 3 (Solvent and Other Product Use)

using the regional growth rates from EDGAR-HYDE. The regional growth rates defined in the RCP historical database are used to extrapolate emissions in categories 1, 2, 4, and 6 back to 1850. Emissions in category 7 are extrapolated using a linear decline to zero in 1850 from the last year with data starting from a 21 year linear trend. In category 3 there are no $CH_4$ emissions reported. The source prioritization and extrapolation is summarized in Table 2.

**N₂O** Country reported data covers 1990 to 2012 for all Annex I countries and some non-Annex I countries. Using EDGAR42 we obtain per country data from 1970 until at least 2010 for all sectors and countries. For agriculture (category 4) the first available year is 1961 from the FAOSTAT dataset and the last year is 2012 for all countries. For the period 1890 to 1970 we use the regional growth rates from the EDGAR-HYDE dataset to extrapolate categories 1, 2, 4, and 6. For the period prior to 1890, the RCP database provides data, but only at a global level and without sectoral detail. We

know of no source that provides regionally or sectorally resolved $N_2O$ emissions prior to 1890. The main contribution to $N_2O$ emissions comes from the agricultural sector, especially the use of manure and nitrogen fertilizers (Davidson (2009)). $N_2O$ emissions are therefore not well correlated with $CO_2$ or $CH_4$ emissions as these have different sources and thus they can not be used as a proxy for $N_2O$ emissions. Data on fertilizer use is only available for a few countries for years earlier than 1961 (Federico (2008)). This is not sufficient for downscaling of agricultural $N_2O$ emissions. We

therefore use the RCP global growth rates which are computed from atmospheric concentration measurements to extend the country time series into the past for all sectors.

**F-gases** Country reported data covers 1990 to 2012 for all Annex I countries and some non-Annex I countries. Other countries are added from EDGAR 42 which also extends existing time series to start in 1970. To extrapolate the data to 1850 we use RCP global growth rates. RCP data and global emissions from EDGAR data are in very good agreement for the time of overlap of the two sources for $SF_6$, HFCs, and PFCs. The time series are obtained using different methods: EDGAR



| Step | Source | Categories | Countries | Years | Type of operation |
|---|---|---|---|---|---|
| 1 | CRF2014 | all | Annex I | 1990 - 2012 | CSG |
| 2 | CRF2013 | all | Annex I | 1990 - 2011 | CSG |
| 3 | UNFCCC2015 | all | 35 non-Annex I | 1990 - 2009 | CSG |
| 4 | BUR2015 | all | 8 non-Annex I | 1994 - 2010 | CSG |
| 5 | FAO2015 | 4 | almost all | 1961 - 2012 | CSG |
| 6 | EDGAR42 | all | almost all | 1970 - 2010 | CSG |
| 7 | EDGAR-HYDE14 | 1, 2, 4, 6 | regions | 1890 - 1995 | growth rates extrap. |
| 8 | RCP | all | global | 1850 - 2005 | growth rates extrap. |
| 9 | numerical | all | all | 1850 - 2014 | linear extrapolation |

**Table 3.** Source prioritization for fossil and industrial $N_2O$. Years are maximal values. Some countries have less coverage. In CRF a few countries have data starting a few years before 1990.

from activity data and emission factors and RCP from inverse emission estimates based on atmospheric concentration measurements. This is a good sign with respect to the uncertainty in the datasets. Because of the similarity in absolute emissions, using RCP growth rates to extend EDGAR data does not significantly alter the global emissions compared to the RCP and is a safe method to obtain emissions back until 1850. Emissions from F-gases are generally very low before
1950 as their large-scale production and use only started in the second half of the 20th century. Technology for large scale production of HFCs was developed in the late 1940s. For PFCs a major breakthrough in industrial production was the Fowler process which was published in 1947 and Industrial production of $SF_6$ began in 1953 (Levin et al. (2010)). The IPCC "Special Report on Safeguarding the Ozone Layer and the Global Climate System" (Metz et al. (2007)) estimated emissions from most HFCs to be zero in 1990 with a steep rise afterwards. However, this is not in agreement with other
sources like EDGAR and RCP which show significant HFC emissions before 1990. As EDGAR and RCP agree on the HFC emissions levels we use the non-zero emissions before 1990. Data for individual F-gases will be provided in a future release of this dataset. Currently this is not possible as some of the sources we use only provide aggregate HFCs and PFCs emissions (UNFCCC2015);

## 4.2 Emissions from land use

The largest share of emissions from land use, land use change, and forestry (LULUCF) is in the form of $CO_2$ originating from deforestation.[10] We therefore focus on $CO_2$ emissions and use a simpler method for $CH_4$ and $N_2O$ emissions. The preparation of the LULUCF pathways follows the same steps as for the fossil fuel and industry pathways. However, due to the high fluctuations in LULUCF data the harmonization of sources is problematic (e.g. when one source shows a sink while

---

[10]The IPCC AR5 WG3 states that " Fluxes resulting directly from anthropogenic FOLU activity are dominated by $CO_2$ fluxes, primarily emissions due to deforestation, but also uptake due to reforestation / regrowth". (Smith et al. (2014))



| Step | Source | Categories | Countries | Years | Type of operation |
|---|---|---|---|---|---|
| 1 | CRF2014 | 2 | Annex I | 1990 - 2012 | CSG |
| 2 | CRF2013 | 2 | Annex I | 1990 - 2011 | CSG |
| 3 | UNFCCC2015 | 2 | 7 non-Annex I | 1990 - 2009 | CSG |
| 4 | BUR2015 | 2 | 2 non-Annex I | 1990 - 2012 | CSG |
| 5 | EDGAR42 | 2 | almost all | 1970 - 2010 | CSG |
| 6 | RCP | 2 | global | 1850 - 2005 | growth rates extrap. |
| 7 | numerical | 2 | all | 1850 - 2014 | linear extrapolation |

**Table 4.** Source prioritization for fluorinated gases. Years are maximal values. Some countries have less coverage. In CRF a few countries have data starting a few years before 1990. F-gas emissions are only reported in category 2. For some countries, data in the BUR and UNFCCC sources is only available for $SF_6$

| Step | Source | Categories | Countries | Years | Type of operation |
|---|---|---|---|---|---|
| 1 | FAOSTAT | 5 | almost all | 1990 - 2010 | copy |
| 2 | Houghton downsc. | 5 | almost all | 1850 - 2005 | copy |
| 3 | numerical | 5 | global | 1850 - 2000 | linear to zero in 1850 |
| 4 | numerical | 5 | all | 1850 - 2014 | linear extrapolation |

**Table 5.** Source prioritization for $CO_2$ from LULUCF. Years are maximal values. Some countries have less coverage.

another source shows emissions for the same period of time). We therefore use the time series from different datasets directly without harmonization. In the preprocessing the Houghton source needs downscaling which is described below.

### 4.2.1 Composition of the land use $CO_2$ pathways

We do not use country reported data for $CO_2$ as there are several ways to calculate anthropogenic land use emissions. Especially
5 developed countries use this freedom to choose an accounting method that results in high $CO_2$ removals which are in contrast to third party sources. We use FAOSTAT data which is available for almost all countries for 1990 to 2012. The period before 1990 is covered by the Houghton dataset which uses 10 regions. In general the period from 1850 to 2005 is covered, but for non-tropical regions the latest years are estimates based on constant extrapolation of the last data point. In some cases this starts as early as 1990. As the period from 1990 on is covered by FAOSTAT data this is no problem for us. In case countries still have
10 missing data we extrapolate into the past using a linear pathway to zero emissions in 1850. Starting point of the extrapolation is a 21 year average. Extrapolation to the future uses a constant derived from the average emissions of the last 15 years. Table 5 summarizes the source creation.



### 4.2.2 Downscaling of HOUGHTON2008

The Houghton source only resolves 10 regions: Canada, China, Europe, Former USSR, Northern Africa and Middle East, Pacific Developed Countries, South and Central America, South and Southeast Asia, Tropical Africa, and the USA. Data for all countries except Canada, China, and the USA has to be computed using downscaling of regional emissions.

As land use emissions are not correlated well with emissions from other sectors we can not use fossil and industrial emissions as a proxy. Instead we use estimates of the conversion of forests into cropland and pasture (deforestation) which is the main source of land use emissions. The methodology we use is based on an approach recently published by Matthews et al. (2014). Estimates of historical deforestation can be computed starting from models of the amount of cropland and pasture required to feed the population in a certain area at a certain time. This time series gives estimates of the land converted to cropland or pasture in that area. Using a dataset of potential natural vegetation (i.e. simulated vegetation in the absence of human interference like deforestation) we compute which fraction of that land was likely covered by forests before the conversion. This gives us a time series of deforested areas on a grid map of the world. The gridded data is transferred into country data using country masks.

The cropland and pasture data is taken from the History Database of the Global Environment (HYDE). We use the SAGE Global Potential Vegetation Dataset to get an estimate of historical forest cover in the absence of human interference. The potential vegetation in this dataset is representative of what vegetation cover would be if anthropogenic interference were removed from the climate and vegetation state observed in the mid-1990s. It therefore does not account for any historic changes in forest area driven by changing climate or atmospheric $CO_2$ concentrations. The SAGE dataset contains 15 separate plant function types (PFTs) of which eight forest/woodland types were combined to generate a simple forest cover mask. The SAGE dataset also includes a PFT for Savanna, which we included in the 'non-forest' category. Although loss of biomass from savanna land has contributed to historical $CO_2$ emissions, we chose to exclude it from this dataset because the carbon density is substantially different to that of forest or woodland areas occurring in the same region. The $CO_2$ emissions downscaling scheme assumes uniform carbon density of vegetation throughout each region, so Savanna was excluded to avoid skewing results. While the different forest PFTs also have different carbon contents, the variability within a region is much smaller than the difference between forest PFTs and savanna within one region. See e.g. Figure 1 of Liu et al. (2015).

The area converted to agricultural land, the sum of cropland and pasture, and that coincides with land that would otherwise be forested is calculated to determine the areal extent of deforestation, and reforestation, over 10 year time steps for each grid cell. Spatial data is converted to country time series using an area-weighted summation according to the country boundaries data of the Food and Agriculture Organization of the United Nations (2015). See also Figure 4.

To downscale the regional emissions data we make the assumption that forests in a region have the same average carbon content. So for any two countries in a region, we assume that converting one hectare of forest into cropland in one country releases the same amount of $CO_2$ to the atmosphere as converting one hectare of forest in the other country. The time resolved data exhibits strong fluctuations which do not necessarily coincide with fluctuations in the emissions data. One reason for this are the different methodological approaches used to create the two datasets. While the Houghton dataset models actual

**Figure 4.** Calculating deforested areas: The two upper plots show the area potentially covered by forests (colored) and the fraction that has been cut until 1850 and 2000 according to the SAGE and HYDE datasets. The third plot shows the difference between the 1850 and 2000 deforestation, thus the area deforested or reforested between 1850 and 2000 which we use to downscale the Houghton dataset.



| Step | Source | Categories | Countries | Years | Type of operation |
|------|--------|-----------|-----------|-------|-------------------|
| 1 | CRF2014 | 5 | Annex I | 1990 - 2012 | copy |
| 2 | CRF2013 | 5 | Annex I | 1990 - 2011 | copy |
| 3 | UNFCCC2015 | 5 | 16 non-Annex I | 1990 - 2009 | copy |
| 4 | BUR2015 | 5 | 3 non-Annex I | 1990 - 2012 | copy |
| 5 | FAOSTAT | 5 | almost all | 1990 - 2012 | copy |
| 6 | EDGAR42 | 5 | almost all | 1970 - 2010 | copy |
| 7 | EDGAR-HYDE14 | 5 | global | 1850 - 2000 | growth rates extrap. |
| 8 | numerical | 5 | global | 1850 - 2000 | linear extrap. (past) |
| 9 | numerical | 5 | all | 1850 - 2014 | linear extrap. (future) |

**Table 6.** Source prioritization for $CH_4$ and $N_2O$ from LULUCF. Years are maximal values. Some countries have less coverage. In CRF a few countries have data starting a few years before 1990.

emissions from deforestation in detail, the method to calculate deforested area uses datasets which are of more theoretical nature. The HYDE dataset models the need for agricultural area in a region and does not represent the agricultural area that was actually present at that time. When population changes, the need for agricultural area changes with it, but the actual agricultural area changes more slowly. This is especially visible in Europe during the second world war. Population and thus

need for agricultural area declined rapidly, leading to afforestation in the model. In reality, agricultural area will remain unused for some time until it is actively afforested or natural vegetation returns and takes up carbon from the atmosphere. This leads to situations where the Houghton source has positive emissions while the SAGE-HYDE calculation show an increase in forest cover indicating $CO_2$ removals. This sign discrepancy causes problems for downscaling (e.g. instability if some countries in a region show afforestation and some deforestation and a general problem of interpreting the shares in afforestation to calculate

shares in deforestation emissions). To solve this problem we do not use yearly shares but cumulative shares in deforestation for the whole period of 1850 to the last data year in the Houghton source to downscale the regional emissions to country level. This approach is also taken in Matthews et al. (2014). Details are given in Appendix D.

### 4.2.3    Composition of the land use $CH_4$ and $N_2O$ pathways

For non-$CO_2$ emissions from land use we use country reported data which is complemented by FAOSTAT and EDGAR42 for

the period from 1970 to 2010. Regional data from EDGAR-HYDE14 is used to extrapolate the time series into the past until 1890 starting from the 1969 value of the 30 year linear trend from 1970 to 1999. For 1850 to 1890 we use a simple linear extrapolation for each gas ($CH_4$, $N_2O$) using the 21 year linear trend of the emissions from 1890 to 1910.

### 4.3    Territorial changes and missing data

We use territorial accounting in this dataset, meaning that emissions that originated from a territory that is now part of country

A are always counted as emissions from country A even if the territory belonged to country B in the year the emissions took



place. However, we can only be as precise as the datasets we are working with. Unfortunately many sources are not very precise with respect to the used methodology. CDIAC $CO_2$ and to a lesser extent FAO data are somewhat of an exception, where split up and merging of countries is made transparent by issuing different country codes. We downscale the data to match the current countries in the way described in Appendix A3. CRF2014, UNFCCC2015 and BUR2015 data are reported by countries and

do not need postprocessing as we use the territorial definitions of the UNFCCC reporting as a basis. For EDGAR data the rules regarding how emissions are assigned to countries in case of territorial changes are not clear from the methodology description and we assume that territorial accounting is used.

For some small countries and countries which became recently independent no emissions data is available yet. In this case we have to construct it using other countries emissions data. Emission data for San Marino and the Vatican are included in

Italian emissions data and downscaled using population shares.[11] This is done on the individual sources during preprocessing (see also Appendix D). For details see Appendix A3. Sudan and South Sudan are also downscaled from emissions of former Sudan using UN population data. (UN Population Division (2015)).

## 5   Data availability

The dataset is available from the GFZ Data Services under the url http://doi.org/10.5880/PIK.2016.003 (Gütschow et al.

(2016)). When using this dataset or one of its updates, please cite this paper and the precise version of the dataset used. Please consider also citing the relevant original sources when using this dataset. Any use of this dataset should also comply with the usage restrictions of the original data sources used for this project.

## 6   Results

In this section we show some key results of our analysis. Details for additional countries, sectors and gases can be explored

on-line on our companion website www.pik-potsdam.de/primap-live/primap-hist/. Here we focus on major emitters and global emissions.

### 6.1   Sectoral distribution of emissions for major emitters

Globally, production and consumption of fossil fuels is responsible for about two thirds of current emissions (Figure 5). An increase from about 50% in 1950 and a negligible contribution in 1850. Before the industrial revolution, deforestation was the

major emissions source followed by agriculture. Currently, these sectors are the second and third largest sources. Roughly 10% of emissions come from waste and industrial processes. Industrial processes increased their share in emissions after 1950 while the share of waste related emissions stayed relatively constant.

The sectoral profile differs strongly among countries (Figure 5). Land use emissions reached almost zero or even negative values in the 1950s to 1970s in industrialized countries (USA, EU, Japan) and a few decades later in China. For all these

---

[11]GDP data is not available.



countries, fossil fuel use and production are the by far largest contributors to total emissions. While the industrialized countries show decreasing (USA, EU) or stagnating (Japan) fossil fuel emissions, China shows rapidly increasing emissions. The increase of emissions from china may have slowed down in the last years, but more time is needed to say if this is more than a temporary effect Korsbakken et al. (2016).

India still has a large share of LULUCF emissions with no clear increase or decrease in the last two decades. Agriculture and LULUCF have similar emissions both in trends and absolute values which have only recently (roughly 1990) been surpassed by the steeply increasing fossil fuel related emissions. For Brazil the largest sector is land use, followed by agriculture. Land use emissions show a decreasing trend, but total emissions do not follow this trend due to a rise in agricultural emissions and fossil fuel related emissions.

Waste gives a small contribution, differing by country without a clear split between developed and developing countries. The contribution of industrial processes is larger in industrialized countries, but especially large in China.

### 6.2   Gas distribution of emissions for major emitters

The contribution of individual gases and gas groups to (GWP weighted) emissions is shown in Figure 6. It is clearly visible that $CO_2$ is by far the largest contributor followed by $CH_4$ and $N_2O$, both globally and for individual countries. The contribution

of fluorinated gases is in general small and negligible for developing countries. Again, China's emissions profile is closer to that of an industrialized country than to other major developing country emitters. Methane emissions are high for countries with a large agricultural sector (India and Brazil). Japan is somewhat of an exception with almost all emissions from $CO_2$.

### 7   Uncertainties

In this paper we do not assess the uncertainties of the dataset in detail. Of the individual datasets used, uncertainty information

is available for some while for others it is not provided. Where it is available, the level of detail is very different. Some datasets give per country or per regional group uncertainty estimates while others only provide global estimates. Individual uncertainty estimates can be over $100\%$. To calculate uncertainty estimates for all countries, gases and sectors for the composite source one has to transform the information given for the individual sources to a common methodology and level of detail and combine it in line with the creation of the composite source. As most datasets come without an uncertainty estimate and third party

estimates are scarce for some datasets it is hard to find a consistent set of uncertainty estimates. Furthermore, different studies use different sectoral resolutions, confidence intervals etc., which makes it difficult to compare and combine the results to arrive at an estimate for our aggregate source. We leave this task for a future publication. In the following, we give a broad overview of the uncertainties of individual sources and compare this dataset to other sources to get an estimate of differences and uncertainties of the source.

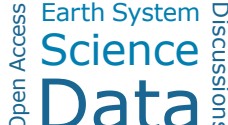



**Figure 5.** Emissions by sector for major emitters and the world. Where land use emissions are negative, the stacked emissions of the other sectors start at this negative value. International shipping and aviation emissions are not included.



**Figure 6.** Emissions by gas for major emitters and the world. International shipping and aviation emissions are not included.





| Country Class | OECD | Europe except OECD | OPEC | Developing countries with stronger statistical bases (e.g. India) | Former USSR and eastern Europe | China and centrally planned Asia | Developing countries with weaker statistical bases (e.g. Mexico) |
|---|---|---|---|---|---|---|---|
| Uncertainty (95% confidence) | 4% | 6.7% | 9.4% | 12.1% | 14.8% | 17.5% | 20.2% |

**Table 7.** Uncertainties for fossil fuel and industrial $CO_2$ emissions for different country groups. All values from Andres et al. (2014)

### 7.0.1 Uncertainties from individual sources

Uncertainty estimates for the CDIAC dataset of global $CO_2$ emissions from fossil fuels and industry have varied since the first assessment made by Marland and Rotty (1984), which resulted in an uncertainty range between 6 and 10% (using a 90% confidence interval). In a recent publication a single global fossil fuel $CO_2$ emission uncertainty of 8.4% (using a 95% confidence interval) is offered as a reasonable combination of data (Andres et al. (2014)), in an attempt to simplify the different assessments and to make the best of the qualitative and quantitative knowledge developed since the first study of 1984.

Different approaches examine CDIAC global uncertainty as the aggregate of the uncertainties associated with fossil fuel $CO_2$ emissions from individual countries. In Andres et al. (1996) several countries of similar perceived uncertainty were grouped together in seven classes. The resulting uncertainty estimates are presented in Table 7.

The EDGAR team has stated that it was not feasible to go beyond the uncertainty tables compiled for EDGAR v2.0, where uncertainties are indicated in terms of ranges ranking from small (10%) to very large (> 100%) (PBL (2010)). However, other institution such as UNEP (UNEP (2012)), estimated an uncertainty range of ±10% (for a 95% confidence interval) for total $CO_2$ (including LULUCF). For global emissions of $CH_4$, $N_2O$, and fluorinated gases uncertainties are estimated to be ±25%, ±30%, and ±20% respectively (using a 95% confidence interval).

FAOSTAT uncertainty estimates are limited to only two carbon pools (above and below-ground biomass) out of six identified by the IPCC guidelines (above and below-ground, dead wood, litter, soil organic carbon, and harvested wood products). Therefore, FAOSTAT estimates of greenhouse gas emissions and removals are likely under-estimated (Federici et al. (2015)). However, Tubiello et al. (2015) provides reasonable overall uncertainty estimates of the FAOSTAT database, where global emission estimates from crop and livestock carry ±30% uncertainty ranges. Uncertainties in the land use sector are even larger, with a ±50% range.

Table 8 gives an overview of available uncertainty estimates for the individual sectors and gases included in the PRIMAP-hist dataset.





| Category | CO$_2$ | CH$_4$ | N$_2$O | FGASES | Kyoto GHG |
|---|---|---|---|---|---|
| 0 | 10%[2] – 20%[5] | 25%[2] – 55%[5] | 30%[2] – 65%[5] | 20%[2] | 30%[5] |
| 0EL | 8.4%[1] – 14%[5] | 55%[5] | 65%[5] | 20%[2] | 25%[5] |
| 1 | 12.5%[5] | 25%[7] | 25%[7] | N/A | 15%[5] |
| 1A | 12.6%[1] | - | - | N/A | - |
| 1B1 | 6%[6] | - | - | N/A | - |
| 1B2 | 6% (25% for 1B2C2)[1] | - | - | N/A | - |
| 2 | 23%[5] | 10%[5] | 50%[5] | 20%[2] | 25%[5] |
| 2A | 23%[1] | 10%[3] | N/A | N/A | - |
| 2B | - | 10%[3] | 50%[3] | N/A | - |
| 2C | - | 10%[3] (2C1) | - | - | - |
| 2D | - | - | - | N/A | - |
| 2E | N/A | N/A | N/A | - | - |
| 2F | N/A | N/A | N/A | - | - |
| 2G | - | - | - | - | - |
| 3 | 10%[7] | N/A | 25%[7] | N/A | 15%[5] |
| 4 | 30%[4] – 100%[3] | 30%[4] – 100%[3] | 30%[4] – 100%[3] | N/A | 65%[5] |
| 5 | 50%[4] | 50%[4] – 75%[3] | 50%[4] – 100%[3] | N/A | 50%[5] |
| 6 | 100%[6] | 100%[3] | 100%[6] | N/A | 100%[5] |
| 7 | 100%[6] | 100%[6] | 100%[6] | N/A | 100%[5] |

**Table 8.** Uncertainties for gases and sectors covered in the PRIMAP-hist dataset. The references are: 1) Andres et al. (2014), 2) UNEP (2012), 3) Olivier et al. (1999), 4) Tubiello et al. (2015), 5) calculated from available data for subsectors and gases, 6) estimated, no data available, 7) Category 0 uncertainty value from 2) used. "N/A" indicates that there are no emission from this gas and sector combination. "-" indicates that we have no uncertainty estimate for the gas - sector combination. Where different uncertainty estimates exist we use the average for the calculation of aggregate uncertainties. Calculations have been carried out according to the IPCC tier 1 methodology. All calculated values are rounded to the nearest multiple of 5% except for CO$_2$ values which are rounded to 1%.

### 7.0.2 Comparison with other data sources

A different approach at uncertainty estimates is to compare different datasets. If they were independent, the distribution of emissions for the same category and gas should represent the uncertainties. This approach also captures uncertainties from different definitions of sectors which are not included in the uncertainties of individual datasets. In Figures 7 and 8 we compare the composite source to some of the individual sources and other composite sources for individual gases and sectors at a global level. To compare the inter-source uncertainty to the individual source uncertainty we also plot an indicative uncertainty range from Table 8 around the PRIMAP-hist dataset. It is apparent that for most categories and gases the inter-source uncertainty is lower than the uncertainty estimated for the individual sources. This means that either the individual uncertainties are overestimated, or that the sources are not independent. Additionally, the number of sources is too small to reliably sample



the 95% confidence interval of the individual source uncertainty. However, the plots show that our composite source is - globally - certainly in agreement with most other sources for almost all sectors and gases within estimated uncertainties. The EDGAR-HYDE data shows relatively low total Kyoto GHG values. The sectors plots show that this is due to low values for industrial processes and land use emissions. The low industrial process emissions can partly be explained by the lack of data for

fluorinated gases in the EDGAR-HYDE dataset, but emissions for $CH_4$ and $CO_2$ are also low. Land use $CO_2$ emissions in the EDGAR-HYDE dataset are only about half of the emissions of all other datasets assessed and outside of the sizable uncertainty range applied to the PRIMAP-hist time series. We have to note that RCP, MATCH, and PRIMAP-hist include HOUGHTON data in their land use time series and are therefore not independent. The HOUGHTON based time series are consistent with EDGAR42 and FAO while the EDGAR-HYDE time series is not similar to any of the time series for more recent emissions.

A further major discrepancy is the RCP $CH_4$ time series, which differs strongly from all other sources. Emissions are significantly higher than in other sources but show a steep decline between 1990 and 2000. No other source used in this analysis shows this effect. RCP $CH_4$ emissions are based on Lamarque et al. (2010) which uses EDGAR-HYDE but adds information for some sectors missing in EDGAR-HYDE14, namely grassland and forest fire emissions.[12] However, the discrepancies can not fully be explained by this as they are present also in other sectors than land use.

For $N_2O$, MATCH and EDGAR42 emissions lower than the PRIMAP-hist dataset while EDGAR-HYDE14 and RCP are higher. MATCH uses EDGAR-HYDE (growth rates) prior to 1990 which explains the very similar pathway profiles and leads to very low emissions outside the uncertainty range before 1970.

   Finally, the estimates of emissions of fluorinated gases are higher for EDGAR42 than for our aggregate dataset. This means that country reported f-gas emissions are significantly lower than what EDGAR calculates the emissions to be.

In conclusion the emissions of this composite dataset globally agree with other sources within uncertainty ranges with a few exceptions where the causes of discrepancies could be explained in most cases.

### 7.0.3 Uncertainties from methodology

The creation of this composite dataset implies several decisions on source prioritization, extrapolations, and downscaling options. These questions usually do not have one "correct" solution but rather different options with individual benefits and

drawbacks. Different options (e.g. linear or constant extrapolation) have different implications for the calculated emissions so the decisions introduce an "expert judgment uncertainty" to the final dataset. A further source of uncertainty is the use of regional growth rates for extrapolation. This assumes that all countries within that region shared the same growth rates which is a simplification. Similarly downscaling uses simplifications like constant emission shares or the use of another source as a proxy. We only use these methods if no individual country data is available and have to accept the uncertainty to fill gaps in

data. See also Section 8 below.

---

[12]International shipping and aviation emissions are also added, but not included in this study.





**Figure 7.** Comparison of the PRIMAP-hist dataset with both individual sources and composite datasets for aggregate Kyoto gases and the main IPCC 1996 categories. Grey shaded areas show the indicative uncertainty range from Table 8 applied to the PRIMAP-hist dataset. Where different uncertainty estimates exist the average value is used. International shipping and aviation emissions are not included.





**Figure 8.** Comparison of the PRIMAP-hist dataset with both individual sources and composite datasets for different gases. Grey shaded areas show the indicative uncertainty range from Table 8 applied to the PRIMAP-hist dataset. Where different uncertainty estimates exist the average value is used. International shipping and aviation emissions are not included.



## 8   Limitations of the method and use of the dataset

When combining time series from different data sources one has to be careful because of the differences in methodology, assumptions, and data underlying the individual sources. The composite source generator of the PRIMAP emissions module was built for this purpose and addresses those problems but some fundamental uncertainties and limitations of the method itself remain. In the following, we explain the sources of data discrepancies and the rationale behind our approach to the generation of a composite source as well as its limitations.

We begin with key sources for uncertainties and differences between datasets.

- Different methodologies for estimating emissions: some datasets are based on end of pipe measurements, others on economic activity data and assumed emission factors. Global emission datasets can also be based on inverse emission estimates from atmospheric concentration measurements.

- Different underlying data: two datasets using the same methodology would have different results when e.g. the data for the electricity production of individual power plants differs. Similarly the data on the exact fuel type used and the emission factors used influence the resulting emissions.

- Differences in the detailed definitions of sectors: there are different ways to categorize emissions by economic sectors and not all data sources use the same categories. Categories from different sources can differ in their exact content despite having broadly the same definition.

- Different assumptions made for variables without data: the uncertainties are especially high for countries without a strong statistical record and sectors and gases which need several assumptions for the calculation of emissions. Power sector $CO_2$ emission have relatively low uncertainty if a good record for power plant technology, the used fuels, and their electricity production exists. Agricultural emissions on the other hand have a high uncertainty as the emissions are based on natural processes which depend on locally and seasonally fluctuating variables like soil moisture (see e.g. Luo et al. (2013)). See also Figure 7.

An overview of the relative uncertainties for the different sources, countries, gases, and sectors is presented in Section 7.

To create a composite dataset we first prioritize the different data sources according to our judgment of their reliability and completeness. More complete sources at the top levels in the hierarchy will create a more consistent dataset than sources which cover only a few sectors or gases. However, if the top-level sources are unreliable, the resulting dataset will be unreliable and it is beneficial to prioritize more reliable but less complete sources. Completeness has different dimensions which we can often not optimize at the same time. Some datasets are very extensive in time and country coverage, but only cover a few gases and sectors (e.g. CDIAC), while other sources cover only a fraction of the countries and years but with almost perfect sectoral and gas resolution (e.g. CRF, UNFCCC, BUR).

The first priority source is used as an anchor point for the other sources which are used to extend the time series and to fill gaps. There are different options for the harmonization needed when extending one source with data from another source. We present some options below, a more detailed discussion is available in Rogelj et al. (2011):



1. no scaling: this does not alter data, but also does not use information from the first priority source to improve data from the lower priority sources.

2. full scaling: here we scale the lower priority sources such that they match the higher priority sources at the borders. Effectively we are using the growth rates of the lower priority sources to extend the higher priority source. If e.g. an emission factor is different for the two sources leading to a large difference in absolute emissions, the growth rates would still be the same and the extension with scaling would effectively use the emissions factor of the first source also for the second source. Of course not all differences come from multiplicative errors like different emission factors. There could also be offsets from missing subsectors or incomplete data on individual emitters which would not be corrected by using growth rates (unless one assumes the same growth rates for the missing subsectors as for the existing sectors).

3. shifting using an offset: the lower priority time series is harmonized by shifting the complete time series by a constant. This method implicitly assumes a constant error over time which is not realistic if the emission time series is not constant. For extrapolation to the past it will likely overestimate emissions while it will likely underestimate emissions for extrapolations to the future (assuming rising emissions).

We use a mixture of approaches 1 and 2. We use scaling but limit the scaling at a factor of 1.5 to avoid introducing additional errors in case of extremely different emissions data.

When combining the different sources we can not take into account all their methodological differences. Often the exact assumptions and underlying data are not published with the datasets and an assessment of the uncertainty of the individual datasets is difficult as useful analysis is scarce (see also previous section). Thus, sometimes a time series using a slightly different sector definition is used to extend another time series. This introduces inconsistencies into the final dataset.

In Section 7 we presented uncertainties of the individual sources, sectors, and gases which can reach over 100% for some gases and sectors. We have to keep that in mind when designing and judging our methods. A very fine tuned and subsector resolved method for the combination of datasets is still bound to the limitations of the input data and their uncertainties. While it is always possible to improve methods to reduce their uncertainty it is not always sensible to invest more time if the major source of uncertainty is the input data and not the processing. Before adding further detail to the PRIMAP-hist dataset it has to assessed if they add real value to the data or are overshadowed by uncertainties of the input data.

When using emissions data one has to respect the uncertainties and limitations of the data in mind. When making a statement about emission intensities in different countries the differences have to be seen in relation to the uncertainties before deducing anything from the calculated values. Individual country uncertainties can be much higher than the global uncertainties presented in Table 8. One of the purposes of this dataset is the calculation of countries contributions to climate change. Again we have to keep uncertainties in mind. This data set can be used to study general effects, such as the impact of pre-1950 emissions on 2100 warming, rather than the exact emission targets for all countries according to a given equity principle (unless one accepts and communicates the uncertainties of the resulting emission targets).

The land use downscaling methodology could be improved by a more detailed treatment of the different plant function types and the inclusion of savannas. For example, the HYDE data does not account for deforestation for firewood which influences




the estimates of deforested areas and the SAGE potential vegetation dataset also removes the human influence on the climate from the simulation. Climate is influenced globally and thus some of the discrepancy between potential and actual vegetation is caused by global climate change and not by local deforestation.[13]

Finally we have to note that the last years are obtained using extrapolations for most sectors and gases. Therefore these data can not be used to make statements about short term emissions trends. We provide a version of this dataset that does not use numerical extrapolation to the future that can be used for this purpose. Where regional data is used for extrapolation to the past individual country developments are not taken into account and can not be deduced from the data. Short term trends can also be influenced by the combination of different sources, thus the consultation of original sources is advised before making statements about such trends.

This dataset is a combination of data from several models, measurements and assumptions including their shortcomings and uncertainties. It adds models and assumptions with new simplifications and uncertainties. However, it gives a more complete picture of the history of countries' greenhouse gas emissions than any of the individual sources can. From this perspective, our aggregate dataset is very useful.

*Author contributions.* All authors contributed to checking the results and writing the manuscript. J.G. conceptualized the study, programmed the necessary addition to the PRIMAP emissions module, created the composite source, and prepared some of the input datasets. L.J. conceptualized and carried out the calculations to obtain deforestation estimates from the SAGE and HYDE datasets. R.Gi. created the accompanying website, most of the figures in this paper, and prepared data. R.Ge. prepared input data. D.S. collected and analyzed the uncertainty data.

*Acknowledgements.* The authors acknowledge and appreciate funding by the Federal Ministry for the Environment, Nature Conservation and Nuclear Safety (11_II_093_Global_A_SIDS_and_LDCs), and the Economic Commission for Latin America and the Caribbean (Development of a reference methodology on historical responsibility for $CO_2$ emissions).

---

[13]Other causes of deforestation are also global e.g. through demand for agricultural products, but under the UNFCCC emissions are attributed to the state they originate from. It is neither considered, where the products are consumed nor where the profits are made.





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

## Appendix A: Details of methodology used

In this section we explain technical details of the methodology used to create this dataset.

### A1    Preprocessing

We use the same methods of preprocessing for all sources, though not all steps are used for all sources. Source specific information is provided in Appendix D.

#### A1.1    Zero data and implausible data

We remove all time series that contain only zero values to ensure that zero values in higher priority sources do not prevent the use of non-zero data from lower priority sources. In case negative data occurs in time series that physically have to be positive we replace the negative data
by zero.

#### A1.2    Gas and category aggregation

Where necessary we aggregate gases to gas baskets (e.g. individual HFCs to the HFCs basket). If data is available at a more detailed sectoral level, we aggregate the categories to obtain time series at the sectoral resolution needed for the PRIMAP-hist dataset. In the process of aggregation we fill gaps in individual time series and extrapolate individual time series such that all gases / subsectors cover the same
time period. Details of the extrapolation methods are discussed in Appendix A5.2 below. The same aggregation routine is also used in postprocessing to aggregate higher categories and the KyotoGHG basket.

### A2    Accounting for territorial changes

Where necessary countries are summed or split to match our territorial definitions. Where only aggregate information is available we use downscaling to obtain country level information. In case we have to downscale emissions of formerly existent larger countries to the current
individual countries, we downscale the larger countries' emissions using constant shares defined by the average of the first five years with data for the individual countries. This is used e.g. for countries of the former USSR. If no data for individual countries is available, we use



an external downscaling key e.g. emissions from a different source or GDP. When countries merge we sum the individual countries. This is used e.g. for Germany.

## A3 Downscaling

We downscale regional data using country shares calculated from a different source, the key. Before downscaling, the key is preprocessed such that time series for all countries present cover the whole period to be downscaled. Extrapolation of country pathways is done using the growth rates of all countries present in the region. This implies that the shares in regional emissions of countries with missing data stay constant from the last year with data (both for extrapolation to the future and to the past). If no data is present for any country in a region for a certain year it is extrapolated using constant emissions implying constant shares for the downscaling. Once the key time series is complete the downscaling itself is done by multiplication of the country shares with the regional data.

## A4 Combination of sources

The main features and functionality of the composite source generator (CSG) are described in Section 4. He we add the missing details. To calculate the harmonization factor to be used for a lower priority source we use the linear trend of the last six years of the higher priority source to calculate a year $n+1$ value (or $n-1$ when extending a time series to the past). The lower priority source is then scaled such that it matches the extrapolated value in the given year. The scaling is confined to the interval $[0.67, 1.5]$ to avoid introducing large changes in emission time series due to scaling.

In case of land use emissions we do not use scaling, but fill gaps with unchanged data from lower priority sources. The high fluctuations of land use data including different signs for data from different sources for the same year introduce high uncertainty in the scaling and render it meaningless in some cases, e.g. when one dataset shows removals while the other shows emissions for the period of overlap.

## A5 Extrapolation

### A5.1 Extrapolation with regional growth rates

For each region in the extrapolation source we loop over all countries contained in the region. We identify if there are years within the given span where the extrapolation source contains data that could extend the country data. If this is the case, we compute the value for the last year without data for the country (the matching year) given by a linear trend. We compute the trend from opposite sides, i.e. for extrapolation to the past from 1850 to 1890 we compute the 1890 value of the country data from a linear trend through 1891 to 1905 and the 1890 value for the regional data from a linear trend through 1876 to 1890. The regional time series is then scaled such that they are identical in the matching year and we extend the country data with the resulting time series. Unless stated otherwise we use 15 year trends.

### A5.2 Numerical extrapolation

In this paper we use numerical extrapolation for extension of time series to the past on the scale of decades where historical data is not available, e.g. for land use $N_2O$ and $CH_4$ emissions. It is also used before the gas and category aggregation process to extrapolate those time series for individual countries, gases and categories which do not have data for the latest years to 2014.

Our framework for numerical extrapolation consists of different methods for extrapolation and a wrapper that controls the results and uses a fall back option if necessary. The following options are available:



**Constant**  Data is extrapolated with a constant value which is computed as the mean of the $n$ last values before the extrapolation. Constant extrapolation has no fall back option.

**Linear**  A linear trend is computed from the last $n$ years before extrapolation. This trend is continued for the period of extrapolation. To control the extrapolated pathway it is checked if it crosses zero (negative emissions are currently impossible for most gases and sectors and have to be excluded). If crossing is not allowed, the fall back option for this case is used. The default option is to replace all values after the crossing point by zero. If emissions are extrapolated to the past and a trend is computed which has higher emissions in the past a fall back option is triggered as well. The default is linear to zero extrapolation.

**Linear to zero**  A linear pathway is constructed from a starting value to zero in the last year of the extrapolation. The starting value is computed from the linear trend of the last $n$ values. If the calculated value is below zero despite all $n$ values being positive, we use the last value instead of the value calculated from the linear trend. There is no fall back option.

**Exponential**  The last $n$ years are used to fit an exponential function which extrapolates the data. A fall back option is used if exponential fitting is not possible (e.g. when the $n$ years contain positive as well as negative values), if too few of the $n$ years have data available, or if during extrapolation to the past we obtain a negative exponent (i.e. emissions in the past higher than in the future). The default fall back option is linear to zero.

Options for all methods are the number $n$ of years to use for the fit (default 15) and the minimal number of these years that have to contain data (default 8). Fitting can be controlled independently for extrapolations to the past and the future.

## Appendix B: Sectoral detail

The sectoral detail of the PRIMAP-hist source for the individual gases is shown in Table 9.

## Appendix C: Territorial definitions

The dataset provides emissions time series for all UNFCCC member states. Some territories are associated to states but have partial independence, other territories claim independence but are not internationally recognized, or have another special status. We include the emissions from these territories in the country emissions if and only if the country includes the emissions when reporting under the UNFCCC. Territories not included in the country reporting are treated independently, however, we can not provide time series for all such territories. Territories which are uninhabited or have only very few inhabitants e.g. in a research station and have no significant emissions are completely excluded from the dataset (Bouvet Island, South Georgia and the South Sandwich Islands). In Table 10 we show which territories are included in countries and which are treated independently and if data is available for territories treated independently. The only territory that is not somehow associated to a single UNFCCC Party is Antarctica. It is included in the dataset despite its negligible anthropogenic greenhouse gas emissions.

In consequence of the Ukraine crisis parts of the (former) Ukrainian territory are currently claimed by both Russia and the Ukraine. The UN has not recognized any changes to the Ukrainian territory so far, so we do not make any adjustments to the Ukrainian emissions. There is no country reported data recent enough to be influenced by the crisis.



| Category | Sector name | Gases |
|---|---|---|
| 0 | National Total | $CO_2$, $CH_4$, $N_2O$, HFCs, PFCs, $SF_6$ |
| 0EL | National Total excluding LULUCF | $CO_2$, $CH_4$, $N_2O$, HFCs, PFCs, $SF_6$ |
| 1 | Total Energy | $CO_2$, $CH_4$, $N_2O$ |
| 1A | Fuel Combustion Activities | $CO_2$, $CH_4$, $N_2O$ |
| 1B1 | Fugitive Emissions from Solid Fuels | $CO_2$ |
| 1B2 | Fugitive Emissions from Oil and Gas | $CO_2$ |
| 2 | Industrial Processes | $CO_2$, $CH_4$, $N_2O$, HFCs, PFCs, $SF_6$ |
| 2A | Mineral Products | $CO_2$ |
| 2B | Chemical Industries | $CO_2$ |
| 2C | Metal Production | $CO_2$ |
| 2D | Other Production | $CO_2$ |
| 2G | Other | $CO_2$ |
| 3 | Solvent and Other Product Use | $CO_2$, $N_2O$ |
| 4 | Agriculture | $CO_2$, $CH_4$, $N_2O$ |
| 5 | Land Use, Land Use Change, and Forestry | $CO_2$, $CH_4$, $N_2O$ |
| 6 | Waste | $CO_2$, $CH_4$, $N_2O$ |
| 7 | Other | $CO_2$, $CH_4$, $N_2O$ |

**Table 9.** Categorical detail of the PRIMAP-hist source for different gases. The categorical hierarchy uses IPCC 1996 terminology. The subcategories of categories 1 and 2 are only resolved for $CO_2$. Other gases are treated at the level of categories 1 and 2. For categories 2E and 2F of the industrial sector there is no data for $CO_2$ because these categories only include the production and consumption of fluorinated gases.

## Appendix D:  Details on data source preprocessing

Here we briefly describe the preprocessing steps carried out for each of the sources used. We only describe the steps for the time series needed for this paper. Aggregation of additional sectors, gas baskets and regional groups is omitted as for this source it is done using the final time series.

5   **BP2015**  BP resolves only some states, other states are summed into five regional groups. we downscale these groups using shares of CDIAC2015 CAT1A emissions. After downscaling countries are summed to the territorial definitions used in this paper.

**BUR2015**  We remove all time series which contain less than three data points or cover less than 11 years. We build the HFCs and PFCs baskets for both SAR and AR4 global warming potentials using the gas and category aggregation functionality of the emissions module (Appendix A1.2). Category aggregation is not necessary as we directly read the data into the PRIMAP emissions database in

10   the needed categorical detail.

**CDIAC2015**  CDIAC makes country unification and splitting explicit by issuing different country codes. We sum and downscale countries where needed to obtain current countries and territories for all years. Where downscaling is needed we use the first 5 years with data for the individual countries as a downscaling key and downscale with constant shares. Where no data for the individual countries is





| Country | countries / territories / dependencies included | countries / territories / dependencies with independent data | countries / territories / dep. without data |
|---|---|---|---|
| Australia | Norfolk Island; Christmas Island; Cocos Islands; Heard and Mc-Donald Islands | | |
| China | | Hong Kong; Macao; Taiwan | |
| Denmark | Faroe Islands; Greenland | | |
| Israel | Palestinian Territory | | |
| France | Saint Barthélemy; Gouadeloupe; French Guiana; Saint Martin; Martinique; Mayotte; New Caledonia; French Polynesia; Reunion; Saint Pierre and Miquelon; Wallis and Futuna; French Southern and Antarctic Lands | | |
| Finland | Åland Islands | | |
| Morocco | Western Sahara | | |
| Netherlands | | Aruba; Netherlands Antilles (Bonaire; Curacau; Saba; Sint Eustatius; Sint Maaten) | |
| New Zealand | | | Tokelau |
| Norway | Svalbard | | |
| United Kingdom | Bermuda; Cayman Islands; Channel Islands; Falkland Islands (Malvinas); Gibraltar; Guernsey; Isle Of Man; Jersey; Montserrat | Anguilla; British Indian Ocean Territory; Pitcairn Islands; Saint Helena, Ascension and Tristan da Cunha; Turks and Caicos Islands; British Virgin Islands | |
| United States | Guam; Northern Mariana Islands; Puerto Rico; American Samoa; US Virgin Islands | | |

**Table 10.** Territorial definitions of countries used in the dataset. The territorial definitions are based on country emission reporting under the UNFCCC and do not imply any political judgment.





available we use CRF2014 data for the same category as downscaling key. This affects downscaling of France and Monaco as well as Switzerland and Liechtenstein. Where CRF data is not available (Italy and San Marino) we use the GDP data from the World Bank (2015) as downscaling key. Finally, we sum countries to the territorial definitions used in this paper.

**CRF2014 and CRF2013** CRF data only needs minimal preprocessing. We build the HFCs and PFCs baskets for both SAR and AR4 global warming potentials using the gas and category aggregation functionality of the emissions module (Appendix A1.2). Actual emissions are used for the PRIMAP-hist dataset (in contrast to potential emissions also available from CRF data).

**EDGAR42** First EDGAR v4.2 and EDGAR v4.2 FT 2010 are independently aggregated to the needed categorical resolution. We retain any existing aggregate time series, as in some cases (at least in EDGAR v4.2 FT2010) not all subsectors are present as individual time series and re-aggregation would loose emissions from the sectors not available individually. Then the two sources are combined using the composite source generator with EDGAR 4.2 FT2010 as the first priority source. The harmonization in the CSG does not use linear trends here. Subsequently HFCs and PFCs gas baskets are aggregated including extrapolation of individual gases such that all gases of a basket cover the same time span. Finally we calculate emissions for some small countries where emissions are included in time series of larger countries. In detail these are: downscaling of Serbia and Montenegro as a region to individual countries, downscaling of Monaco from France, downscaling Liechtenstein from Switzerland, and downscaling of Vatican and San Marino from Italy. The downscaling key used is population data from the UN Population Division (2015).

**EDGAR-HYDE14** EDGAR-HYDE data uses the EDGAR v2.0 categorization which differs from the IPCC 1996 categorization used here. The IPCC 1996 categories we identify with the EDGAR42 categories are shown in Table 11. The summation of subcategories is done

| EDGAR-HYDE | IPCC1996 |
| --- | --- |
| FNN | CAT1A |
| FPP | CAT1B |
| I00 | CAT2 |
| LGG + LNN + L42 + L43 + L70 | CAT4 |
| L41 | CAT5 |
| W10 + WNN | CAT6 |

**Table 11.** Category matching for EDGAR-HYDE and IPCC 1996 categories.

using the emissions module's aggregation framework. We do not use international bunker fuel emissions (EXX) as we do not include bunker fuels in this analysis. Data is interpolated using Matlab's 'pchip' function.

**FAO2015** Like CDIAC, FAO data explicitly models splitup and unification of countries. Our first step is to sum and split these countries to obtain time series for the current countries and the territorial definitions used here (see Section C).FAO uses different subcategories for agriculture and land use than IPCC 1996 which need to be translated to IPCC 1996 categories. For this paper the details are not relevant as we operate on aggregate agricultural and land use data.

**HOUGHTON2008** The downscaling is described in Section 4.2.2. Here we add some details. The downscaling uses regional shares in cumulative deforested areas to split the regional emissions pathway to countries. In some regions there are both countries with net deforestation and net afforestation, so some countries have negative shares which can not be used directly for downscaling in a meaningful way. Instead we first calculate shares from only deforestation and multiply those with the regional pathway to obtain



preliminary emission pathways. These pathways are then shifted such that the cumulative net emissions (or removals) equal the cumulative net emissions (or removals) calculated directly from the net deforestation shares. This approach avoids inverted growth rates for countries with net afforestation in a region with net deforestation.

Countries missing in the Houghton source are added using the regional growth rates and shares computed by the relative deforestation compared to a Houghton region with similar climate.

**HYDE** No preprocessing is needed.

**RCP** Data is first interpolated using MATLAB's 'pchip' function. For $CH_4$ we aggregate time series to the necessary regional level. HFCs and PFCs baskets are created. For $CH_4$ from categories 1, 2, and 4 the years $1860 - 1880$ are removed before interpolation. They show a steep decline to almost zero emissions from 1850 to 1860 which rise again to much higher values in 1890. This can not be observed in the data presented in Lamarque et al. (2010) which is the original source of the data according to the RCP website (Meinshausen (2011)). We judge this to be an error and thus replace the values by interpolation.

**SAGE** No preprocessing is needed.

**UNFCCC2015** See BUR2015.

### Appendix E: Data sources not used

In this section we describe data sources that were considered but not used in the final composite source and give the reasons why the data was not used.

### E1 Biennial Reports

Biennial Reports are submitted to the UNFCCC by Annex I Parties. The UNFCCC biennial reporting guidelines for developed country Parties (Decision 2.CP17, Annex I) state that "the information provided in the biennial report should be consistent with that provided in the most recent annual inventory submission, and any differences should be fully explained". It is therefore safe to assume that data submitted with the BRs is consistent with CRF data (Section 2.2.3).

### E2 National communications by developed countries

National communications by developed country Parties UNFCCC (2014b) serve the purpose to give information on the commitments Parties are undertaking to limit their greenhouse gas emissions and the policies implemented and planned to reach the commitments. They contain some greenhouse gas data but historical data does not add to CRF data, so national communications by developed country Parties are not used here.

### E3 CAIT 2.0

The Climate Analysis Indicators Tool (CAIT) dataset is published by the World Resources Institute (WRI). It contains data for several countries until 2011 (some countries have less coverage). Emission time series are available either on an aggregate Kyoto GHG level, or with details for either sectors or gases. Unfortunately there is no data with details for sector and gas at the same time. For F-gases only aggregate data is available without the details on HFCs, PFCs and SF6 needed for this project.



Similar to our work, CAIT 2.0 emissions time series are assembled from different sources. Data from the International Energy Agency (IEA) (see Appendix E6), the U.S. Energy Information Administration (EIA) (see Appendix E5), and CDIAC (see Section 2.3.2) are used for fossil $CO_2$. Non-$CO_2$ emissions are taken from the US EPA source (see Appendix E7). LULUCF data are taken from FAO (see Section 2.3.4).

All sources are either included in our dataset individually (CDIAC, FAOSTAT), not publicly accessible (IEA), or only contain emissions already covered from other sources (EIA, USEPA). We do not use CAIT data, as the results are more transparent when using the original data sources directly.

## E4    CDIAC $CH_4$

This dataset has been described in Stern and Kaufmann (1995, 1996, 1998) and covers global $CH_4$ emissions for a period from 1860 to
1994. It is created using correlations of methane emissions to socioeconomic variables or emissions of other gases for which time series are available. It is tested against emission estimates from measurements of atmospheric methane concentrations. Due to its lack of country or regional data it could only be used for extrapolation. However, we have RCP data that covers the same period and sectoral detail but has a regional resolution. we therefore do not use the CDIAC CH4 data.

## E5    EIA Energy $CO_2$

The U.S. Energy Information Administration's (EIA) publishes $CO_2$ emissions from energy consumption for most of the world countries. The period from 1980 to 2012 is covered. The covered sectors are consumption of coal, petroleum, and natural gas (together these correspond to IPCC 1996 category 1A) and flaring of natural gas (IPCC 1996 category 1B2C22).

We do not use the dataset because the sectors and time frame are covered by CDIAC2015.

## E6    IEA Energy $CO_2$

The International Energy Agency offers $CO_2$ emissions from fuel combustion for purchase. The dataset covers 34 OECD countries and 100 non-OECD countries. As it is not publicly available we do not include it in our dataset.

## E7    USEPA

The United States Environmental Protection Agency (EPA) published data for non-$CO_2$ emissions (US Environmental Protection Agency (2012)). The dataset covers many countries and the years 1990 to 2005. It is a composite of different data sources where publicly-available
country-prepared reports are prioritized. A main source for the historical data is the UNFCCC flexible query system. Annex I countries therefore use CRF data while non-Annex I countries use data from the National Communications and National Inventory Reports. However, each time series has only a few data points. We already include the individual sources used in this dataset and only little information is added. Thus we do not use the USEPA data.