# Peer review of "The PRIMAP-hist national historical emissions time series"

_Earth System Science Data, 2016_

## Referee Comment (RC1) · Anonymous Referee #1 · 2 Jul 2016

The paper "The PRIMAP-hist national historical emission time series" describes the emission data set for each country and the gases covered by Kyoto protocol. The data set is based on the combination of the published data sets. The added value as well as possible applications of the composite inventory are not clearly articulated in the paper. The assessment is biased to the period from 1990, while historical emissions are not covered properly. The composite data set is created as a pure statistical exercise through series of multiplications, additions and extrapolations, through it is not clear if the data sets can be combined together, under what condition, and what the final combination actually means. The final product is expressed through IPCC 1996 main categories, but is unclear how the diverse categories used in original inventories were converted into IPCC categories. Language of the paper also needs correction. The paper is full of jargon and abbreviations, as well as punctuation mistakes. In places

explanations are extremely brief and do not allow to judge if the sources of information are appropriate for combination or not. The paper is also unspecific in many places in relation to what gases and what sectors are addressed (everything is just emissions). More specific comments: p.1 (abstract) – what is a temporal resolution of the composite inventory p.2 (10-15) – to what extent current IPCC emission categories are applicable to historical emissions (as there was a substantial change in technology, use of fuels, efficiency etc.)? p.3 (20) – reference is made to the use of growth rates. Are these emissions growth rates (at what temporal resolution)? How those are constructed? p.4 – CDIAC is presented as a country specific inventory. How the historical boarders are taken into consideration in this inventory? p.5 (7) – please spell out KP p.7 – sections 2.3.1 and 2.3.2 are very brief. How do these pieces complement each other or do they provide a repetitive information. Do both sources use the same emission factors for their calculations? How different are the estimates based on fossil fuel consumption and fossil fuel burning and flaring? Section 2.3.2 refers to the emissions based on the statistics of fossil fuel production and trade combined with the information on the chemical composition (of what?) for the period 1751 to 1949. Where does such information come from? p.7 (26-30) – the paragraph claims that emissions in EDGAR are calculated using technology dependent emission factors for EACH technology determined by the end of pipe measurements. If each emission factors could have been determined as claimed, the emission inventories would have been much more precise. There are plenty of technologies, especially in developing countries, which are very poorly described in a sense of emission factors, the same applies to historical emission factors. p.8 (5)- The sentence states that land use emissions are categorized into forestland, grassland, cropland and biomass burning, but they are applicable to $CO_2$ only. It is a bit confusing though, as cropland should theoretically include rice paddies, which are one of the important source of $CH_4$. Line 10 later speaks of agricultural emissions. What is the difference between cropland land use emissions and agricultural emissions? How these emission categories are mapped into IPCC categories? p.8 – sections 2.4.1 and 2.4.2 are extremely short. How these two sources compare

to each other? Do they use the same regional definitions and source categories? If not, how the original data were processed for the composite inventory? Line 21 refers to constant emissions outside of tropical regions prior to 1990. Where do these data come from? Based on what principle the HOUGHTON2008 is downscaled to country level? Description of RCP historical dataset is very poor. What gases are covered? In what cases atmospheric inverse modelling is used? p.9 (7) – the need for categories mapping is indicated, but details of how it is done are not provided p.9 – sections 2.6.1 and 2.6.2 have very short description of data sources. What emission categories and gases are covered? How do these inventories compare with the one of FAO? p.11 presents the source preprocessing but does not describe how diverse categories are mapped against IPCC before being used in the composite source generator p.13 – the compilation of the composite source as presented on the figures creates a question of consistency. If the inventories are different during one period and you just combine them multiplying by coefficients, what value does it bring? What additional uncertainty is introduced through such data treatment? p.15 (15-16) – how far back can the RCP growth rates based on atmospheric concentration measurements be extended? Is this approach applicable to all countries and all sectors? p.17 (10) – does the statement mean that you assume no land use emissions before 1850? p.18 – how large forest fires are reflected in the considered historical emissions? p.21 (23) – what emissions are discussed here? The paragraph is absolutely unclear. p.22 (27) – it is very unfortunate that the uncertainty estimate of the dataset is left for a future publication as uncertainty is a natural part of any dataset delivered and allow for evaluation of applicability of the data set for specific tasks. p.26 – section 7.0.2 presents the comparison of the compiled product with the other data sources, though the comparison is made with the data sources used to create the composite inventory. It is obvious that as it is a composite product of original datasets it will follow the features the authors prioritize in compilation, hence such a comparison does not bring any value and cannot be considered as an independent verification.

---

## Referee Comment (RC2) · Anonymous Referee #2 · 10 Jul 2016

Review ESSD-2016-12

Overall, this seems like a careful and constructive effort toward a very useful product. I applaud the authors for thinking to make this data publicly available and for the substantial effort to provide a reliable and well-documented description.

The manuscript as submitted suffers from some intellectual and compositional sloppiness. For one glaring example, without searching separately this reader still would have no idea what the acronym PRIMAP refers to?

Several of the Figures have serious presentation deficiencies that render them almost useless. These deficiencies subtract substantially from the overall quality of the product. Find a communication officer at PIK and use their help to change, revise and overall improve the graphics.

The overall discussion - very necessary - on data uncertainties seems to present serious logic failures. See detailed comments below.

Page 1, line 24: We need definition of the EDGAR acronym here, where it first appears?

Page 2, line 10: We have no definition of the acronym PRIMAP?

Page 2, lines 13,14: "For details and sector names we refer to Table 1 9." Confusing! This does not seem to point to Table 1 on page 14 but rather to Table 9 in Appendix B. Rather than referring the reader to an end-of-the document appendix, why not put the Table directly here where readers would find it useful?

Page 2, lines 29,30: "In the supplementary information we present a list of territories included in the emissions of UNFCCC 30 Parties as well as information on the territories that are treated separately (Section C)." Again confusing. 'Supplementary information' to this reader implies a supplement but this paper has no supplement and in any case ESSD and Copernicus do not archive supplements. Does these refer to Table 10 in Appendix C? Again, why not put it here if useful, or at least provide a hot link.

Page 3, line 15: "… preprocessing is available in Section D." This refers to Appendix D?

Page 5, line 8: The authors defined KP (Kyoto Protocol) in a footnote but not yet in the main text?

Page 5, line 11: which convention?

Page 6, line 27: Here we read that, under UNFCCC CRF, the 2015 edition used IPCC 2006 categories. But a few lines earlier (line 15) we read about IPCC 2006 categorisation as a disqualifying factor. Do we need more-exact definitions of the IPCC 2006 categories? Do we have a terminology problem? We need clarification. From subsequent text, e.g. lines 2 and 3 on page 7, we get the impression that 2006 IPCC categories represent a future and desirable standard not yet implemented in this data set? Clarification needed! Perhaps Table 9, which in the caption refers to IPCC 1996, provides some hints? If so, we need a stronger declaration here, we should NOT need to look at Table 9 and guess.

Page 7, line 15: "United Nations energy statistics" these presumable come from a different source than United Nations FCCC but from where? Do we get a reference or we need to go to the source materials?

Page 7, line 20: "… as well as other substances." Other regulated substances?

Page 7, line 22, 23:  Confusing use of punctuation in this section.  Presumably Unep means UNEP?

Page 8, line 1: FAOSTAT - does this source report annual data?

Page 8, line 22: "past 1990".  Prior to 1990?  Since 1990?  Figure 1 implies that one can use Houghton et al. data back to 1850?

Page 8, line 24: implies that CMIP5 comes from IPCC but in fact it comes from World Climate Research Programme?

Page 8, line 28: give explicit credit for the MAGICC6 chemistry model (NCAR etc.)?

Page 10, line 24: Although referenced together in this sentence, Table 6 occurs several pages after Table 5.

Page 12, Figure 2:  A useful but graphically fairly simple figure.  Could the authors redraw it to show exactly the sequence used by the CSG for this data product?

Page 13, Figure 3:  We need much more information and explanations about the steps, the sequence and the labels (or absence of labels) to find this figure useful!

Page 14: Presumably one could match the Table 1 steps to Figure 3.  Table 1 caption could contain additional useful information?  E.g column categories refers to IPCC 1996 categories? Likewise for Tables 2, 3, and 4?

Page 15, lines 5 to 16: A link to Table 3 in this section?

Page 16, lines 1 to 13: A link to Table 4 in this section?

Page 23, Figure 5: axis labels, category labels, panel labels all unreadable or missing.  Need a substantial revision of this figure and with a substantial caption!  Likewise for Figure 6, page 24!!

Page 21 and 22, Section 6: This section has no impact or utility because of the severe deficiencies in Figure 5 and Figure 6!

Page 25, line 9: Does Table 7 come from Andres 2014 or Andres 1996?  Confusing.

Pages 28, 29, Figures 7 and 8:  Again basically useless due to lack of panel labels, series labels, axis labels.  Did no one look at these figures before submission?

Pages 22 to 27, the entire Section 7: (Why do we have sub-section 7.0.1, 7.0.2, etc?) The authors correctly describe the extreme challenges of extracting, compiling and reporting composite uncertainty estimates.  Table 8 illustrates the problem to the extreme. But then the authors fail to follow their own advice and cautions and build this section largely - to the extent we can only guess from the unreadable Figures 7 and 8 - on comparison of the generated data set to the individual source data included in the composite.  They know this is invalid, they say this is invalid, then they do it anyway!  The entire section needs revision and rewriting to clarify how much they can not conclude because of this lack of independence of source and product and how much - if anything - they conclude without violating this fundamental requirement.

Section 8: This reader found Section 8 much more realistic and cautious, even if less quantitative, than section 7.

Appendices:  Some of this information could go more usefully directly in the text.  Only Appendix A, D and E seem useful in separate form.

---

## Short Comment (SC1) · 11 Jul 2016

We thank Anonymous Referee #2 for bringing the incorrect display of some figures in the manuscript to our attention. It appears they only display wrongly in some PDF readers and platforms.

We will work with ESSD to make sure the final manuscript displays correctly for all readers and have attached a zip file with separate PDFs of the figures.

All further comments by Anonymous Referees #1 and #2 we will address in the coming days.

[Figure]

Please also note the supplement to this comment:
http://www.earth-syst-sci-data-discuss.net/essd-2016-12/essd-2016-12-SC1-supplement.zip
* * *

---

## Author Comment (AC1) · 9 Sep 2016

September 9, 2016

Dear Anonymous Referee #1,

thank you for your detailed comments. Here we try to address the concerns you
expressed in your report. We start with some general proposals for restructuring the
paper to make the information contained more accessible and address the individual
comments thereafter.

**A.  General remarks and restructuring of the manuscript**

Several comments are asking for more information on individual data sources and their
processing. This information is often given in the appendix of the paper, so it seems
that the information in the appendix is not well accessible. We thus merged parts of
the appendix into the main text (Appendices B, C) and improved references to the
other sections of the Appendix (A, D, E). All references made in this document are to
the original Appendix numbering.

   Below we answer each individual comment and explain which changes we made to
the manuscript.

**B.  Answers to general comments**

**Comment 1** *The paper "The PRIMAP-hist national historical emission time series"*
*describes the emission data set for each country and the gases covered by Kyoto pro-*
*tocol. The data set is based on the combination of the published data sets. The added*
*value as well as possible applications of the composite inventory are not clearly artic-*
*ulated in the paper.*

**Answer 1.** The added value of this dataset compared to the individual datasets used
here is its completeness in terms of countries, sectors and time which no existing

dataset can deliver. Furthermore, the methodology has been improved compared to older composite datasets (as explained in the introduction of the manuscript).

Previous versions of the PRIMAP-hist dataset have been used in the UNEP gap report 2015 and the INDC factsheets published by the Australian-German Climate and Energy College (http://www.climate-energy-college.net/indc-factsheets). Predecessors of the dataset, especially the PRIMAP4 baseline [1] have been used e.g. for the Climate Action Tracker[2] and the publication "National post-2020 greenhouse gas targets and diversity-aware leadership" by Meinshausen et al., 2016.

**Change to manuscript 1.** We have added information on the added value and applications and have provided examples of use of previous versions of the PRIMAP-hist dataset and its predecessors.

**Comment 2** *The assessment is biased to the period from 1990, while historical emissions are not covered properly.*

**Answer 2.** From 1990 on there is more detailed data available, especially data reported by countries and expert reviewed (for developed countries). We use the best available data for each period.

**Change to manuscript 2.** No changes to manuscript.

**Comment 3** *The composite data set is created as a pure statistical exercise through series of multiplications, additions and extrapolations, through it is not clear if the data sets can be combined together, under what condition, and what the final combination actually means.*

**Answer 3.** We explain the caveats of combining different datasets in the paper, especially in Section 8. The aim of the paper is to create a comprehensive inventory of historical greenhouse gas emissions. The combination of existing emissions sources is our way to achieve this. The methods we use do have underlying models and assumptions which are explained in Sections 4.1 and 8. We also refer to the existing literature on these methods.

**Change to manuscript 3.** The description of the underlying models in Section 4.1 has been improved.

**Comment 4** *The final product is expressed through IPCC 1996 main categories, but is unclear how the diverse categories used in original inventories were converted into IPCC categories.*
* * *
[1] https://www.pik-potsdam.de/research/climate-impacts-and-vulnerabilities/research/rd2-flagship-projects/primap/emissionsmoduledocumentation/primap-baseline-reference

[2] www.climateactiontracker.org

**Answer 4.** Most sources we use in the paper are published using IPCC 1996 categories: CRF2014/2013, UNFCCC2015, BUR2015, RCP, EDGAR42(FT2010). For EDGARHYDE we lay out the mapping in Appendix D. The FAOSTAT database uses different subcategories to the IPCC 1996 categories, but the main categories (agriculture and land use) coincide. CDIAC2015 and BP2015 use categories that have a direct counterpart in the IPCC 1996 categorization and don't need mapping.

**Change to manuscript 4.** We have added the information on the IPCC categories to the dataset descriptions for all sources which are not published in IPCC 1996 categories.

**Comment 5** *Language of the paper also needs correction. The paper is full of jargon and abbreviations, as well as punctuation mistakes.*

**Answer 5.** Thank you for pointing this out.

**Change to manuscript 5.** We have checked that all abbreviations are introduced properly and proof read for language again.

**Comment 6** *In places explanations are extremely brief and do not allow to judge if the sources of information are appropriate for combination or not.*

**Answer 6.** Thank you for pointing this out.

**Change to manuscript 6.** We have added to the source descriptions based on the detailed comments.

**Comment 7** *The paper is also unspecific in many places in relation to what gases and what sectors are addressed (everything is just emissions).*

**Answer 7.** Thank you for pointing this out.

**Change to manuscript 7.** We have made the language more specific based on the detailed comments.

**C. Answers to more specific comments**

**Comment 8** *p.1 (abstract) – what is a temporal resolution of the composite inventory*

**Answer 8.** The composite dataset uses a yearly resolution for all time periods.

**Change to manuscript 8.** The information has been added to the abstract.

**Comment 9** *p.2 (10-15) – to what extent current IPCC emission categories are applicable to historical emissions (as there was a substantial change in technology, use of fuels, efficiency etc.)?*

**Answer 9.** The categories itself are applicable for all time periods, though the detailed categories were created with current technologies in mind. Where categories refer to specific technologies, no emissions are reported before this technology was available (e.g. for the categories tailored towards the use and production of fluorinated gases). However, the technology specific categories are not on the upper levels on the category tree that is used for the PRIMAP-hist dataset. An exception being categories 1 and 2 where we do use subcategories for $CO_2$. For category 1 the subcategories we use are technology independent as the distinction is between emissions from burning of fossil fuels and fugitive emissions which are again split into two subcategories based on the fuel types. For category 2 most subcategories are again technology independent and based on industrial sectors like chemical industries or mineral products. Technological improvements change emissions in these categories, but the categories itself are applicable independent of the technologies used.

**Change to manuscript 9.** No changes to manuscript.

**Comment 10** *p.3 (20) – reference is made to the use of growth rates. Are these emissions growth rates (at what temporal resolution)? How those are constructed?*

**Answer 10.** The use of growth rates is explained in more detail later in the paper in Section 4.1 and Appendices A4 and A5.1.

**Change to manuscript 10.** We added details and a reference to Section 4.1 and Appendices A4 and A5.1.

**Comment 11** *p.4 – CDIAC is presented as a country specific inventory. How the historical boarders are taken into consideration in this inventory?*

**Answer 11.** The CDIAC publication from Andres et al., Carbon dioxide emissions from fossil-fuel use, 1751–1950 (Tellus, 1999) [3] states the following: "*Almost all political entities represented in the pre-1950 production and trade data were already represented in the modern CDIAC time series. The major exception was a united Korea, under Japanese hegemony, which existed from 1905 to 1944 and was assigned the discrete country code of 409. Land exchanges between countries were also accommodated, when possible. For example, the emissions from Alsace-Lorraine were included with Germany or France, reflecting which political unit governed these lands at any given time. This maintained the integrity of political entities despite changes in national borders.*" Where CDIAC issues different country codes for countries that merged or split apart we sum or downscale the CDIAC data to obtain data for the countries currently existing.
* * *
[3]http://www.tellusb.net/index.php/tellusb/article/view/16483

**Change to manuscript 11.** We have included the pieces of information which are not present already in our description of the emissions accounting in the PRIMAP-hist dataset.

**Comment 12** *p.5 (7) – please spell out KP*

**Answer 12.** Thank you for noting this.

**Change to manuscript 12.** We have spelled out KP at its first occurrence.

**Comment 13** *p.7 – sections 2.3.1 and 2.3.2 are very brief. How do these pieces complement each other or do they provide a repetitive information. Do both sources use the same emission factors for their calculations? How different are the estimates based on fossil fuel consumption and fossil fuel burning and flaring? Section 2.3.2 refers to the emissions based on the statistics of fossil fuel production and trade combined with the information on the chemical composition (of what?) for the period 1751 to 1949. Where does such information come from?*

**Answer 13.** The two datasets described in Sections 2.3.1 and 2.3.2 contain information for similar sectors. CDIAC covers more sectors than BP as it also provides $CO_2$ emissions from flaring and cement production while BP only provides energy related fossil fuel burning emissions. However, the BP time series is more up to date and provides information for the latest years that is not contained in the CDIAC time series. Therefore BP complements CDIAC and we use both sources. It is generally the case that several sources provide information for the same sectors and years, but none of the sources is complete in terms of sectors, gases, countries, and years.

**Change to manuscript 13.** We have added information on why we use BP additionally to CDIAC and updated the CDIAC source description to provide missing information.

**Comment 14** *p.7 (26-30) – the paragraph claims that emissions in EDGAR are calculated using technology dependent emission factors for EACH technology determined by the end of pipe measurements. If each emission factors could have been determined as claimed, the emission inventories would have been much more precise. There are plenty of technologies, especially in developing countries, which are very poorly described in a sense of emission factors, the same applies to historical emission factors.*

**Answer 14.** "*end of pipe* **measurements**" was actually a typo, correctly it should say "*end of pipe* **measures**". Thank you for pointing to that.

The EDGAR methodology [4] states that "*Emissions (EM) for a country C are calculated for each compound x on an annual basis (y) and sector wise (for i sectors, multiplying on the one hand the country-specific activity data (AD), quantifying the human activity for each of the i sectors, with the mix of j technologies (TECH) for each sector i, and with their abatement percentage by one of the k end-of-pipe (EOP) measures for each technology j, and on the other hand the country-specific emission factor (EF) for each sector i and technology j with relative reduction (RED) of the uncontrolled emission by installed abatement measure k, as summarized in the following formula:*

$$EM_C(y,x) =$$
$$\sum_{i,j,k} [AD_{C,j} \cdot TECH_{C,i,j}(y) \cdot EOP_{C,i,j,k}(y) \cdot EF_{C,i,j}(y,x) \cdot (1 - RED_{C,i,j,k}(y,x))] \text{ "}$$

The EDGAR inventory as well as some other inventories (CRF, UNFCCC) offer a greater level of detail than used for the PRIMAP-hist dataset. However, to combine the different inventories we have to use the least common denominator and use the resolution of the datasets with less sectoral resolution. Especially for historical emissions before 1970 there is no data available which offers significantly more detail than the main IPCC 1996 categories. The uncertainties for emissions in the individual categories of the EDGAR dataset are substantial so we do not see a contradiction to your (certainly correct) assessment that a lot of the technologies are poorly described by emission factors.

**Change to manuscript 14.** The typo has been corrected.

**Comment 15** *p.8 (5)- The sentence states that land use emissions are categorized into forestland, grassland, cropland and biomass burning, but they are applicable to CO2 only. It is a bit confusing though, as cropland should theoretically include rice paddies, which are one of the important source of CH4. Line 10 later speaks of agricultural emissions. What is the difference between cropland land use emissions and agricultural emissions? How these emission categories are mapped into IPCC categories?*

**Answer 15.** The land use emissions do not cover the emissions directly introduced by agricultural use, but emissions from soil changes that are caused by agricultural use of the soil. FAOSTAT states that "*Greenhouse gas (GHG) emissions data from cropland are currently limited to emissions from cropland organic soils. They are those associated with carbon losses from drained histosols under cropland.*"[5]. The emissions from the agricultural use itself are covered under the agriculture sector where rice cultivation is a subsector[6]. As stated in Appendix D the aggregate land use and agricultural sectors we work with in this paper coincide with IPCC 1996 sectors so no mapping is required for this paper.
* * *
[4]http://edgar.jrc.ec.europa.eu/methodology.php
[5]http://faostat3.fao.org/modules/faostat-download-js/PDF/EN/GC.pdf
[6]http://faostat3.fao.org/browse/G1/GR/E

**Change to manuscript 15.** We have added information to make the definitions of the land use and agricultural sectors more clear.

**Comment 16** *p.8 – sections 2.4.1 and 2.4.2 are extremely short. How these two sources compare to each other? Do they use the same regional definitions and source categories? If not, how the original data were processed for the composite inventory? Line 21 refers to constant emissions outside of tropical regions prior to 1990. Where do these data come from? Based on what principle the HOUGHTON2008 is downscaled to country level? Description of RCP historical dataset is very poor. What gases are covered? In what cases atmospheric inverse modelling is used?*

**Answer 16.** The Houghton dataset described in Section 2.4.1 contains only $CO_2$ emissions from land cover change. The RCP dataset described in Section 2.4.2 contains emissions for all Kyoto gases and sectors. However, the land use $CO_2$ emissions of the RCP dataset are not used for PRIMAP-hist as they are based on the Houghton dataset (and regional detail is lost).

The constant emissions outside tropical regions in the Houghton dataset are obtained using the assumption that emissions calculated for 1990 are also valid for the subsequent years by the original authors of the dataset. We do not use the extrapolated data in the PRIMAP-hist dataset. The downscaling of the Houghton data is described in Section 4.2.2.

For more detailed information on the RCP historical emissions please also consult the paper "The RCP greenhouse gas concentrations and their extensions from 1765 to 2300"[7] and the RCP scenario data group website[8].

**Change to manuscript 16.** Information has been added to Sections 2.4.1 and 2.4.2.

**Comment 17** *p.9 (7) – the need for categories mapping is indicated, but details of how it is done are not provided*

**Answer 17.** Details are provided in Appendix D.

**Change to manuscript 17.** A reference to Appendix D has been added.

**Comment 18** *p.9 – sections 2.6.1 and 2.6.2 have very short description of data sources. What emission categories and gases are covered? How do these inventories compare with the one of FAO?*

**Answer 18.** The datasets covered in these sections do not contain any emissions data. Instead SAGE contains data for potential vegetation and HYDE contains simulation data of past real vegetation. By comparing those we can determine areas where deforestation has occurred which we use to downscale the Houghton data to country level. More information on the use of these datasets is provided in Section 4.2.2.
* * *
[7]Meinshausen, M., Smith, S.J., Calvin, K. et al. Climatic Change (2011) 109: 213. doi:10.1007/s10584-011-0156-z

[8]http://www.pik-potsdam.de/ mmalte/rcps/

**Change to manuscript 18.** We have added an introduction to Section 2.6 to clarify that the gridded datasets we use do not contain emissions data.

**Comment 19** *p.11 presents the source preprocessing but does not describe how diverse categories are mapped against IPCC before being used in the composite source generator*

**Answer 19.** Please see answer to Comment 4.

**Change to manuscript 19.** Please see answer to Comment 4. Additionally we have added a reference to the source specific preprocessing which includes category mapping.

**Comment 20** *p.13 – the compilation of the composite source as presented on the figures creates a question of consistency. If the inventories are different during one period and you just combine them multiplying by coefficients, what value does it bring? What additional uncertainty is introduced through such data treatment?*

**Answer 20.** The method uses the year to year growth rates of the lower priority source to extend the higher priority sources. The prioritization of the sources takes the completeness and reliability of the absolute values into account to use the most reliable absolute values and the year by year growth rates of the other sources to extend those data. A similar method is employed in the Global Carbon Project.[9] The uncertainty of the final time series can be calculated using the error propagation formulas. For a scaling $f$ the standard deviation of a scaled time series $C = f \cdot B$ would be $s_c = \sqrt{s_f^2 + s_B^2}$, where $s_B$ is the standard deviation of the time series $B$ and $s_f$ is the standard deviation of the scaling factor which depends on $s_B$ and $s_A$ in a manner determined by the exact matching algorithm ($A$ denotes the time series which $B$ is adjusted to).

**Change to manuscript 20.** We have added information on the uncertainty propagation of this operation to the uncertainty section.

**Comment 21** *p.15 (15-16) – how far back can the RCP growth rates based on atmospheric concentration measurements be extended? Is this approach applicable to all countries and all sectors?*

**Answer 21.** We use the RCP data from 1850 forward to the 20th century with the exact end date differing by gas, category, and country. We do not have to extend RCP data as it already covers the wanted time period. The RCP dataset only contains global emissions on an economy wide level. It is a simplification to use the same growth rates for all countries and sectors, however, there is no other dataset available, so the only other option would be to use numerical extrapolation or to leave a gap in the dataset which we find worse than using global growth rates.
* * *
[9]Le Quéré, C., Moriarty, R., Andrew, R. M., Canadell, J. G., Sitch, S., Korsbakken, J. I., . . . Zeng, N. (2015). Global Carbon Budget 2015. Earth System Science Data, 7(2), 349–396. http://doi.org/10.5194/essd-7-349-2015

**Change to manuscript 21.** No changes to manuscript.

**Comment 22** *p.17 (10) – does the statement mean that you assume no land use emissions before 1850?*

**Answer 22.** We do not consider emissions before 1850 in this paper, but implicitly we assume no emissions before 1850 when setting the 1850 value to zero. We use 0 in 1850 to rather under- than overestimate emissions when extrapolating. Using growth rates is difficult for land use because of the strong fluctuations. This extrapolation is just used for very few small countries. Pre-1850 land use emissions have a small effect on cumulative emissions and accounting for them would "results in a shift of attribution of global temperature increase from the industrialized countries to less industrialized countries, in particular South Asia and China, by up to 2–3%" according to Pongratz And Caldeira (2012)[10]. Pre-industrial land use change emissions could be included in a future version of this dataset.

**Change to manuscript 22.** Information on the impact of pre-industrial land use emissions has been added. A list of countries which are affected by the linear extrapolation has been added.

**Comment 23** *p.18 – how large forest fires are reflected in the considered historical emissions?*

**Answer 23.** The publications describing the dataset states that only direct (deliberate) human induced activities are covered.[11][12] Thus generally, forest fires are not included except for fire clearing. However, for the USA wildfires and the effect of measures for fire suppression are included.

**Change to manuscript 23.** The information has been added.

**Comment 24** *p.21 (23) – what emissions are discussed here? The paragraph is absolutely unclear.*

**Answer 24.** All references to "emissions" in this sectors shall be read as references to "emissions of aggregate Kyoto greenhouse gases".

**Change to manuscript 24.** We have clarified the language in the section.
* * *
[10] Pongratz, J., and Caldeira, K. (2012). Attribution of Atmospheric CO2 and Temperature Increases to Regions: Importance of Preindustrial Land Use Change. Environmental Research Letters, 7(3), 34001. http://doi.org/10.1088/1748-9326/7/3/034001

[11] HOUGHTON, R. A. (2003), Revised estimates of the annual net flux of carbon to the atmosphere from changes in land use and land management 1850–2000. Tellus B, 55: 378–390. doi:10.1034/j.1600-0889.2003.01450.x

[12] Houghton, R. A. (1999). The annual net flux of carbon to the atmosphere from changes in land use 1850–1990*. Tellus B, 298–313. doi:10.1034/j.1600-0889.1999.00013.x

**Comment 25** *p.22 (27) – it is very unfortunate that the uncertainty estimate of the dataset is left for a future publication as uncertainty is a natural part of any dataset delivered and allow for evaluation of applicability of the data set for specific tasks.*

**Answer 25.** As explained in Section 7 there is a gross lack of uncertainty data for the individual datasets. As long as uncertainty data for the individual datasets is not available, for a detailed uncertainty analysis one would need to make several assumptions on the uncertainty of emission factors etc. Without detailed knowledge and understanding of the methodology underlying the individual datasets this is a new source of uncertainty. This is not feasible to do for all the datasets used in this paper in the scope of this paper.

**Change to manuscript 25.** No changes to manuscript.

**Comment 26** *p.26 – section 7.0.2 presents the comparison of the compiled product with the other data sources, though the comparison is made with the data sources used to create the composite inventory. It is obvious that as it is a composite product of original datasets it will follow the features the authors prioritize in compilation, hence such a comparison does not bring any value and cannot be considered as an independent verification.*

**Answer 26.** The comparison is not intended as an independent verification but to show how the composite sources compares to the individual source and older composite sources. A comparison of the composite source with the individual sources is meaningful, as we only use the year to year growth rates of the lower priority sources such that the absolute values of the composite source are governed by the highest priority source.

**Change to manuscript 26.** We have improved the section to clarify the goal and legitimacy of the comparison and to make sure that we don't say more than intended to and possible from the data.

best regards
Johannes Gütschow on behalf of all co-authors

---

## Author Comment (AC2) · 9 Sep 2016

September 9, 2016

Dear Anonymous Referee #2,

thank you for your detailed comments. Here we try to address the concerns you
expressed in your report. We start with some general proposals for restructuring the
paper to make the information contained more accessible and address the individual
comments thereafter.

**A.  General remarks and proposals for restructuring**

Several comments (especially from Referee #1) are asking for more information on
individual data sources and their processing. This information is often given in the
appendix of the paper, so it seems that the information in the appendix is not well
accessible. We thus merged parts of the appendix into the main text (Appendices
B, C) and improved references to the other sections of the Appendix (A, D, E). All
references made in this document are to the original Appendix numbering.

Below we answer each individual comment and explain which changes we made to
the manuscript.

**B.  Answers to general comments**

**Comment 1** *Overall, this seems like a careful and constructive effort toward a very
useful product. I applaud the authors for thinking to make this data publicly available
and for the substantial effort to provide a reliable and well-documented description.*

*The manuscript as submitted suffers from some intellectual and compositional slop-
piness. For one glaring example, without searching separately this reader still would
have no idea what the acronym PRIMAP refers to?*

**Answer 1.** Thank you for this comment.

**Change to Manuscript 1.** Changes to manuscript are explained in answers to detailed comments.

**Comment 2** *Several of the Figures have serious presentation deficiencies that render them almost useless. These deficiencies subtract substantially from the overall quality of the product. Find a communication officer at PIK and use their help to change, revise and overall improve the graphics.*

**Answer 2.** Thank you for pointing this out. The presentation deficiencies of the figure are display issues. The figures display correctly in the submitted pdf, however, after the ESSDD header is added almost all text disappears in some pdf viewers. We have submitted corrected figures to the discussion page of the paper and will work with the publisher to make sure the display issues are fixed for the revised version of the manuscript.

**Change to Manuscript 2.** Figures are correct in the revised version of the manuscript.

**Comment 3** *The overall discussion - very necessary - on data uncertainties seems to present serious logic failures. See detailed comments below.*

**Answer 3.** See answers to detailed comments below.

**Change to Manuscript 3.** See changes for detailed comments below.

**C. Answers to more specific comments**

**Comment 4** *Page 1, line 24: We need definition of the EDGAR acronym here, where it first appears?*

**Answer 4.** Thank you for pointing this out.

**Change to Manuscript 4.** The definition of the acronym has been added.

**Comment 5** *Page 2, line 10: We have no definition of the acronym PRIMAP?*

**Answer 5.** PRIMAP is the acronym for *Potsdam Real-time Integrated Model for probabilistic Assessment of emissions Paths* and the name of the research group at the Potsdam Institute for Climate Impact Research which mainly authored this study.

**Change to Manuscript 5.** The definition of the acronym has been added.

**Comment 6** *Page 2, lines 13,14: "For details and sector names we refer to Table 1 9." Confusing! This does not seem to point to Table 1 on page 14 but rather to Table 9 in Appendix B. Rather than referring the reader to an end-of-the document appendix, why not put the Table directly here where readers would find it useful?*

**Answer 6.** Thank you for pointing out the wrong reference and the suggestion to add the table to the main text. We placed it in the appendix because we assumed only few readers would look at the details but we are happy to change this.

**Change to Manuscript 6.** The table has been moved to the main text.

**Comment 7** *Page 2, lines 29,30: "In the supplementary information we present a list of territories included in the emissions of UNFCCC 30 Parties as well as information on the territories that are treated separately (Section C)." Again confusing. 'Supplementary information' to this reader implies a supplement but this paper has no supplement and in any case ESSD and Copernicus do not archive supplements. Does these refer to Table 10 in Appendix C? Again, why not put it here if useful, or at least provide a hot link.*

**Answer 7.** A former version of the manuscript was split into the main text and supplementary information instead of an appendix. The reference has not been adjusted when switching from supplementary information to an appendix. The reference is indeed to Appendix C.

**Change to Manuscript 7.** We have integrated Appendix C into the main text.

**Comment 8** *Page 3, line 15: "… preprocessing is available in Section D." This refers to Appendix D?*

**Answer 8.** Yes.

**Change to Manuscript 8.** The reference has been corrected.

**Comment 9** *Page 5, line 8: The authors defined KP (Kyoto Protocol) in a footnote but not yet in the main text?*

**Answer 9.** Thank you for pointing that out.

**Change to Manuscript 9.** KP is spelled out at its first occurrence.

**Comment 10** *Page 5, line 11: which convention?*

**Answer 10.** The United Nation Framework Convention on Climate Change (UNFCCC).

**Change to Manuscript 10.** "Under the convention" has been removed from the sentence as the preceeding sentence already references the UNFCCC.

**Comment 11** *Page 6, line 27: Here we read that, under UNFCCC CRF, the 2015 edition used IPCC 2006 categories. But a few lines earlier (line 15) we read about IPCC 2006 categorisation as a disqualifying factor. Do we need more-exact definitions of the IPCC 2006 categories? Do we have a terminology problem? We need clarification. From subsequent text, e.g. lines 2 and 3 on page 7, we get the impression that 2006 IPCC categories represent a future and desirable standard not yet implemented in this data set? Clarification needed! Perhaps Table 9, which in the caption refers to IPCC 1996, provides some hints? If so, we need a stronger declaration here, we should NOT need to look at Table 9 and guess.*

**Answer 11.** As you pointed out it is desirable to move to IPCC 2006 categories. However, so far only few data sources are available in this categorization and even those are not complete yet. UNFCCC CRF data, 2015 edition was not available for all countries by the time the manuscript has been submitted. Of the Biennial Update Reports only very few were submitted using IPCC 2006 categories. It is possible to transform datasets from 1996 to 2006 categories, but we did not want to do that without even one relatively complete dataset in 2006 categories. The next update of the dataset will use updated CRF data and IPCC 2006 categories.

**Change to Manuscript 11.** We have added information on the reasons for using IPCC 1996 categories.

**Comment 12** *Page 7, line 15: "United Nations energy statistics" these presumable come from a different source than United Nations FCCC but from where? Do we get a reference or we need to go to the source materials?*

**Answer 12.** The CDIAC website[1] states that the UN Energy Statistics Yearbook[2] is used.

**Change to Manuscript 12.** The reference has been added.

**Comment 13** *Page 7, line 20: "... as well as other substances." Other regulated substances?*

**Answer 13.** Some of the other substances are controlled by the Montreal Protocol (e.g. HCFCs), others are not controlled (e.g, black carbon, organic carbon)

**Change to Manuscript 13.** This information has been added to the manuscript.

**Comment 14** *Page 7, line 22, 23: Confusing use of punctuation in this section. Presumably Unep means UNEP?*
* * *
[1] http://cdiac.ornl.gov/trends/emis/overview_2013.html
[2] United Nations. 2016. 2013 Energy Statistics Yearbook. United Nations Department for Economic and Social Information and Policy Analysis, Statistics Division, New York.

**Answer 14.** Thank you for pointing that out. Unep means UNEP, the Bibtex style changed the case automatically.

**Change to Manuscript 14.** We have adjusted the punctuation and moved the references to the ends of the individual sentences to make the text less confusing. The case problem in the UNEP reference has been fixed.

**Comment 15** *Page 8, line 1: FAOSTAT - does this source report annual data?*

**Answer 15.** Yes.

**Change to Manuscript 15.** The information has been added.

**Comment 16** *Page 8, line 22: "past 1990". Prior to 1990? Since 1990? Figure 1 implies that one can use Houghton et al. data back to 1850?*

**Answer 16.** We mean "since 1990". Houghton data can be and is used back to 1850.

**Change to Manuscript 16.** We have clarified the "past 1990" statement and added the information that the time series starts in 1850.

**Comment 17** *Page 8, line 24: implies that CMIP5 comes from IPCC but in fact it comes from World Climate Research Programme?*

**Answer 17.** Thank you for pointing this error out.

**Change to Manuscript 17.** The error has been corrected.

**Comment 18** *Page 8, line 28: give explicit credit for the MAGICC6 chemistry model (NCAR etc.)?*

**Answer 18.** Good idea.

**Change to Manuscript 18.** Explicit credit is given.

**Comment 19** *Page 10, line 24: Although referenced together in this sentence, Table 6 occurs several pages after Table 5.*

**Answer 19.** You are right that this is unfortunate. Tables 5 and 6 are placed in the sections with the detailed information on the creation of the time series for the respective gases and sectors.

**Change to Manuscript 19.** We have adjusted the reference on page 10. The tables are kept where they are in the respective sections with detailed source creation information.

**Comment 20** *Page 12, Figure 2: A useful but graphically fairly simple figure. Could the authors redraw it to show exactly the sequence used by the CSG for this data product?*

**Answer 20.** The steps used in this dataset differ per gas and sector. They are summarized in the Tables 1–5. Figure 2 shows the detail for an individual CSG step.

**Change to Manuscript 20.** We will add to the caption of Figure 2 to better explain how it relates to the source creation in our case and add references to the relevant sections and to Figure 3 which gives the details for the creation of one individual time series.

**Comment 21** *Page 13, Figure 3: We need much more information and explanations about the steps, the sequence and the labels (or absence of labels) to find this figure useful!*

**Answer 21.** Explanation of the steps is given in text in Section 4.1 and Section A of the Appendix. The labels and titles are present in the original plots and will be present in the revised manuscript.

**Change to Manuscript 21.** The above mentioned references have been added to the figure caption.

**Comment 22** *Page 14: Presumably one could match the Table 1 steps to Figure 3. Table 1 caption could contain additional useful information? E.g column categories refers to IPCC 1996 categories? Likewise for Tables 2, 3, and 4?*

**Answer 22.** Yes, the steps in Table 1 correspond to the steps in Figure 3.

**Change to Manuscript 22.** We have emphasized the connection between Table 1 and Figure 3 and added information to the captions of Tables 1–5.

**Comment 23** *Page 15, lines 5 to 16: A link to Table 3 in this section?*

**Answer 23.** That link should definitely be there.

**Change to Manuscript 23.** The link has been added.

**Comment 24** *Page 16, lines 1 to 13: A link to Table 4 in this section?*

**Answer 24.** This link should also be there.

**Change to Manuscript 24.** The link has been added.

**Comment 25** *Page 23, Figure 5: axis labels, category labels, panel labels all unreadable or missing. Need a substantial revision of this figure and with a substantial caption! Likewise for Figure 6, page 24!!*

**Answer 25.** The labels and titles are present in the original plots and will be present in the revised manuscript. See also Comment 2.

**Change to Manuscript 25.** See Comment 2. The figure captions have been improved with references to the relevant sections.

**Comment 26** *Page 21 and 22, Section 6: This section has no impact or utility because of the severe deficiencies in Figure 5 and Figure 6!*

**Answer 26.** We believe that with the correct figures it does have impact and utility. See also Comment 2.

**Change to Manuscript 26.** See Comment 2.

**Comment 27** *Page 25, line 9: Does Table 7 come from Andres 2014 or Andres 1996? Confusing.*

**Answer 27.** The tables comes from Andres 2014, however, the grouping was introduced in Andres 1996.

**Change to Manuscript 27.** We have clarified where the grouping and the table come from.

**Comment 28** *Pages 28, 29, Figures 7 and 8: Again basically useless due to lack of panel labels, series labels, axis labels. Did no one look at these figures before submission?*

**Answer 28.** See Comment 2.

**Change to Manuscript 28.** See Comment 2.

**Comment 29** *Pages 22 to 27, the entire Section 7: (Why do we have sub-section 7.0.1, 7.0.2, etc?) The authors correctly describe the extreme challenges of extracting, compiling and reporting composite uncertainty estimates. Table 8 illustrates the problem to the extreme. But then the authors fail to follow their own advice and cautions and build this section largely - to the extent we can only guess from the unreadable Figures 7 and 8 - on comparison of the generated data set to the individual source data included in the composite. They know this is invalid, they say this is invalid, then they do it anyway! The entire section needs revision and rewriting to clarify how much they can not conclude because of this lack of independence of source and product and how much - if anything - they conclude without violating this fundamental requirement.*

**Answer 29.** The subsection numbering is a mistake. Thank you for pointing out that our conclusions seem to be too far fetched. It is a good point that the comparison reads like a validation of our data which it can not be because of the interdependencies between sources and the lack of detailed uncertainty data. Our intention was to show that the data source does not contradict most other sources given the large uncertainty estimates.

**Change to Manuscript 29.** The subsection numbering has been fixed. The section has been revised to make sure it does not say more than intended and justifiable from the data.

**Comment 30** *Section 8: This reader found Section 8 much more realistic and cautious, even if less quantitative, than section 7.*

**Answer 30.** Thank you for your comment.

**Change to Manuscript 30.** No changes to manuscript.

**Comment 31** *Appendices: Some of this information could go more usefully directly in the text. Only Appendix A, D and E seem useful in separate form.*

**Answer 31.** Thank you for the suggestion. It is always hard to judge which details the reader finds interesting and which not.

**Change to Manuscript 31.** We have integrated Appendices B and C into the main text and improved references to Appendices A, D, and E.

best regards
Johannes Gütschow on behalf of all co-authors

---

## Author Comment (AC3) · 13 Sep 2016

**The PRIMAP-hist national historical emissions time series**

Johannes Gütschow[1], M. Louise Jeffery[1], Robert Gieseke[1], Ronja Gebel[1], David Stevens[1],
Mario Krapp[2], and Marcia Rocha[2]

[1]Potsdam Institute for Climate Impact Research, Telegraphenberg A 31, 14473 Potsdam, Germany
[2]Climate Analytics, Friedrichstraße 231, Haus B, 10969 Berlin, Germany

*Correspondence to:* J. Gütschow (johannes.guetschow(at)pik-potsdam.de)

**Abstract.** To assess the history of greenhouse gas emissions and individual countries' contributions to emissions and climate change, detailed historical data is needed. We combine several published datasets to create a comprehensive set of emission pathways of each country and Kyoto gas covering the years 1850 to 2014 with yearly values for all UNFCCC member states as well as most non-UNFCCC territories. The sectoral resolution is that of the main IPCC 1996 categories. Additional subsectors are available for time series of $CO_2$ from energy and industry. Country resolved data is combined from different sources and supplemented using year to year growth rates from region resolved sources and numerical extrapolations to complete the dataset. Regional deforestation emissions are downscaled to country level using estimates of the deforested area obtained from potential vegetation and simulations of agricultural land. In this paper, we discuss the data sources and methods used and present the resulting dataset including its limitations and uncertainties. The dataset is available from http://doi.org/10.5880/PIK.2016.003 and can be viewed on the website accompanying this paper (www.pik-potsdam.de/primap-live/primap-hist/).

**1 Introduction**

The question of responsibility for climate change and its impacts plays a significant role in the UNFCCC[1] negotiations  around the global agreement to limit the global mean temperature increase and avoid dangerous climate change. It is interlinked with the discussion about equitable access to sustainable development, which forms the basis of different frameworks to assess if climate targets put forward by countries reflect a "fair share" in the collective burden to reshape the economy towards emissions neutrality. The Brazilian delegation to the UNFCCC has put forward a framework that assesses a country's contribution to climate change by calculating the fraction of the total warming generated by that country's historical greenhouse gas emissions. This approach is explained in Miguez and Filho (2000) and has been quantified in Höhne et al. (2010); Elzen et al. (2013); Matthews et al. (2014) amongst others. Other effort sharing proposals use cumulative per capita emissions as a metric and thus also need a detailed record of historical emissions by individual countries (Winkler et al. (2011); Baer et al. (2008); Bode (2004)). In 2001 the MATCH[2] expert group was set up by the UNFCCC to generate historical emissions time series for this purpose. The dataset which resulted from this effort proved very useful in the negotiations and the scientific
* * *
[1]United Nations Framework Convention on Climate Change

[2]Ad hoc group for the modeling and assessment of contributions of climate change, www.match-info.net

community (Höhne et al. (2010)). It was updated in Elzen et al. (2013) with  data from the Emissions Database for Global Atmospheric Research v4.2 (EDGAR) data to cover all gases and emission until 2010. The Climate Analysis indicator Tool (CAIT) also publishes a historical greenhouse gas emissions data set, which is a composite of other sources. However, non-$CO_2$ emissions are only covered for recent years (1990 - 2012) and it resolves either sectors or gases but not both at
5  the same time. The CAIT dataset is a composite of different sources, most of them are also included in the dataset presented here. The Global Carbon Project publishes the Global Carbon Budget (Le Quéré et al. (2015) ), which covers the atmospheric concentration of $CO_2$ and its sources and sinks. The fossil fuel $CO_2$ emissions data used is taken directly from other sources; non-$CO_2$ emissions data is not included.

Here we present a historical emissions dataset with a finer sectoral resolution, newly available input data, and new and
10  improved methods for the combination of datasets. Previous versions of the PRIMAP-hist (PRIMAP - Potsdam Real-time Integrated Model for probabilistic Assessment of emissions Paths) dataset have been used in the UNEP gap report 2015 (UNEP (2015) ) and the INDC factsheets published by the Australian-German Climate and Energy College (Meinshausen and Alexander (2 Predecessors of the dataset, especially the PRIMAP4 baseline[3] have been used e.g. for the Climate Action Tracker[4] and in Meinshausen et al. (2015) . The dataset presented here has been improved in categorical resolution, time coverage, and country
15  coverage compared to its predecessors. Methodological improvements include extapolation with regional growth rates, more sophisticated downscaling methods (e.g. for land use emissions), and category and gas aggregation that automatically inter- and extrapolates missing data.

We build our time series from a range of publicly available data sources (see Section 2), which are prioritized based on their completeness and reliability – an approach that has also been taken by the IPCC to compile the historical dataset for the 5th
20  Assessment Report (IPCC (2014), Annex.II.9, Historical data). For each time series (country, gas, sector resolved) the lower priority sources are used as  year by year growth rates[5] to extend the higher priority sources. Where no country data is available we use regional growth rates, growth rates from  superordinate sectors, and numerical extrapolation to complete the time series.

For land use emissions we use the approach introduced in Matthews et al. (2014) and downscale a regional dataset using
25  estimates of deforested areas derived from simulations of potential vegetation and agricultural land.

The PRIMAP-hist dataset covers the six Kyoto greenhouse gases and gas groups. Independent time series are generated for carbon dioxide ($CO_2$), methane ($CH_4$), nitrous oxide ($N_2O$), hydrofluorocarbons (HFCs), perflurocarbons (PFCs) and sulfur hexafluoride ($SF_6$). For all gases except $CO_2$ the sectoral resolution is that of the main IPCC 1996 categories. For $CO_2$ more detailed categories are used because some important datasets cover only subsectors of categories 1 and 2. For details and sector
30  names we refer to Table  1. $NF_3$ is not included as it has only been included into the group of Kyoto Protocol relevant gases for the second commitment period of the Kyoto Protocol, which started in 2013 and data availability is therefore still
* * *
[3]https://www.pik-potsdam.de/research/climate-impacts-and-vulnerabilities/research/rd2-flagship-projects/primap/emissionsmoduledocumentation/primap-baseline-refere
[4]www.climateactiontracker.org
[5]Other publications use the term "rate of change".

| Category | Sector name | Gases |
|---|---|---|
| 0 | National Total | $CO_2$, $CH_4$, $N_2O$, HFCs, PFCs, $SF_6$ |
| 0EL | National Total excluding LULUCF | $CO_2$, $CH_4$, $N_2O$, HFCs, PFCs, $SF_6$ |
| 1 | Total Energy | $CO_2$, $CH_4$, $N_2O$ |
| 1A | Fuel Combustion Activities | $CO_2$, $CH_4$, $N_2O$ |
| 1B1 | Fugitive Emissions from Solid Fuels | $CO_2$ |
| 1B2 | Fugitive Emissions from Oil and Gas | $CO_2$ |
| 2 | Industrial Processes | $CO_2$, $CH_4$, $N_2O$, HFCs, PFCs, $SF_6$ |
| 2A | Mineral Products | $CO_2$ |
| 2B | Chemical Industries | $CO_2$ |
| 2C | Metal Production | $CO_2$ |
| 2D | Other Production | $CO_2$ |
| 2G | Other | $CO_2$ |
| 3 | Solvent and Other Product Use | $CO_2$, $N_2O$ |
| 4 | Agriculture | $CO_2$, $CH_4$, $N_2O$ |
| 5 | Land Use, Land Use Change, and Forestry | $CO_2$, $CH_4$, $N_2O$ |
| 6 | Waste | $CO_2$, $CH_4$, $N_2O$ |
| 7 | Other | $CO_2$, $CH_4$, $N_2O$ |

**Table 1.** Categorical detail of the PRIMAP-hist source for different gases. The categorical hierarchy uses IPCC 1996 terminology. The subcategories of categories 1 and 2 are only resolved for $CO_2$. Other gases are treated at the level of categories 1 and 2. For categories 2E and 2F of the industrial sector there is no data for $CO_2$ because these categories only include the production and consumption of fluorinated gases.

scarce. In the remainder of the manuscript we use the term fluorinated gases to refer to the combined group of gases HFCs, PFCs, and $SF_6$.

We use the IPCC 1996 categories instead of the new IPCC 2006 categories, because almost all data sources are reported using the 1996 categories and we can avoid conversions between categorizations by using the 1996 categories. The UNFCCC is switching towards IPCC 2006 categories for data reported by countries, however, issues with the reporting software resulted in some countries delaying their emissions reporting and others asked the UNFCCC not to display the reported data. We plan to switch to the IPCC 2006 categories for a future release of the PRIMAP-hist dataset once these problems are solved. The emissions time series cover the period of 1850 to 2014. This is achieved through the combination of various sources and extrapolation for some sectors, gases, and countries both into the past and into the future. The extent of the extrapolation needed varies between sectors, gases and countries. Data coverage is very good for energy related $CO_2$ emissions for the whole period. For other gases and sectors we have to rely on growth rates from regional data for the period before 1970. The data sources we use are described in Section 2, while the details of the combination process, including the prioritization, are described in Section 4.

The time series starts in 1850 for all sectors including land use. Pre-1850 land use emissions have a small effect on cumulative emissions and accounting for them would "*results in a shift of attribution of global temperature increase from the industrialized countries to less industrialized countries, in particular South Asia and China, by up to 2–3%*" (Pongratz and Caldeira (2012) ). On the other hand, uncertainties are especially high for early emissions, which limits the usefulness of the additional data. However, pre-industrial land use change emissions could be included in a future version of this dataset.

[revised manuscript text omitted]

Appendix B contains some additional information on the creation of the emissions pathways with individual fluorinated gases combined together.

20 ## 2.3 Country resolved data

**2.3.1 BP Statistical Review of World Energy [BP2015]**

The BP Statistical Review of  World Energy is published every year and contains time series for $CO_2$ emissions from consumption of oil, gas, and coal (which corresponds to IPCC 1996 category 1A). Emission data are derived on the basis of the carbon content of the fuels and statistics of fuel consumption. The 2015 edition (British Petroleum (2015)) contains information 25 for 76 individual countries and 5 regional groups of smaller countries, which we downscale to country level. The first year in the time series is 1965, the last is 2014. Appendix B gives details on the downscaling. We use the BP data additionally to sources covering similar gases and categories (e.g. CDIAC) because it offers emissions data for recent years, which is not included in the other sources.
* * *
[10]When we write "all" there can still be a few exceptions where data is missing for single countries or sectors.

[11]No 2015 CRF data has been submitted by Belarus and Switzerland. Canada and the United States have submitted data but requested to not make it publicly available until problems with the CRF reporter software are solved.

**2.3.2 CDIAC fossil $CO_2$ [CDIAC2015]**

The CDIAC fossil fuel and industrial $CO_2$ emissions dataset is published by the Carbon Dioxide Information Analysis Center (CDIAC) with regular updates (Boden et al. (2015)). It covers emissions from fossil fuel burning, flaring, and cement production for 221 countries and territories. The first year is 1751 and the last year 2011. Emissions from 1751 to 1949 are computed using statistics of fossil fuel production and trade combined with information on the chemical composition of the fuels and assumptions on the use and combustion efficiency following the methodology presented in Andres et al. (1999). Emission data for the years 1950 to 2011 are based primarily on the United Nations  Energy Statistics Yearbook (United Nations (2016) ) using the methodology presented in Marland and Rotty (1984).

The data requires some preprocessing to account for split up and unification of countries. The preprocessing methodology and mapping of emissions categories are explained in Appendix B.

**2.3.3 EDGAR versions 4.2 and 4.2 FT2010 [EDGAR42]**

The EDGAR (Emissions Database for Global Atmospheric Research) dataset is published by the European Commission Joint Research Center (JRC) and Netherlands Environmental Assessment Agency (PBL). It undergoes regular updates. The current version is 4.2. It contains emissions data for all Kyoto greenhouse gases as well as other substances.[12] It covers 233 countries and territories in all parts of the world, though not all countries have full data coverage. EDGAR version 4.2  covers the period 1970 to  2008 (JRC and PBL (2011) ). Additionally the EDGAR v4.2 FT2010  covers the period 2000 to 2010  (JRC and PBL (2013); Olivier and Jans EDGAR v4.2 FT2012  covers 1970 to 2012 but only for $CO_2$, $CH_4$, $N_2O$, and aggregate KyotoGHG emissions with no sectoral resolution (JRC and PBL (2014); UNEP (2014) ). Version 4.3 covering the period until 2012 has been implicitly announced but is not yet available (as of February 1st 2016).[13]

EDGAR time series are calculated using activity data on a per sector, per gas and per country basis. Emissions are calculated using a country, sector, and gas specific technology mix with technology dependent emission factors. The emission factors for each technology are determined by end of pipe measurements, country specific factors, and a relative emission reduction factor to incorporate installed emissions reduction technologies.

Appendix B contains information on the combination of EDGAR v4.2 and EDGAR v4.2 FT2010, as well as details on the category and gas basket aggregation and country preprocessing.
* * *
[12]Some of the other substances in the EDGAR database are controlled under the Montreal Protocol (HCFCs), others are not controlled so far (e.g. black carbon, organic carbon)

[13]In the "Trends in Global $CO_2$ emissions report" (Olivier et al. (2015)) EDGAR v4.3 is referenced as forthcoming in 2015.

**2.3.4 FAOSTAT database [FAO2015]**

The Food and Agriculture Organization of the United Nations (FAO) publishes data  with yearly values for emissions from agriculture and land use (Food and Agriculture Organization of the United Nations (2015a)). Over 200 countries and territories are included in the database.

5      The land use emissions are categorized into forestland, grassland, cropland, and biomass burning where the first three categories contain information on $CO_2$ only, while biomass burning also contains information on $N_2O$ and $CH_4$ emissions. To generate the time series, data from land use and forestry databases (both from FAO and other institutions) are used together with IPCC estimates of emission factors and the FAO "Global Forest Resources Assessment" database for carbon stock in forest biomass. For details we refer to the methodology information on the FAOSTAT website (Food and Agriculture Organization

10 of the United Nations (2015b)). The time series cover 1990 to 2012.

The land use emissions do not cover the emissions directly introduced by agriculture, but emissions from soil changes that are caused by agricultural use of the soil. For cropland FAOSTAT states that "*Greenhouse gas (GHG) emissions data from cropland are currently limited to emissions from cropland organic soils. They are those associated with carbon losses from drained histosols under cropland.*" (Food and Agriculture Organization of the United Nations (2016) ).

15      FAOSTAT data for agricultural emissions ranges from 1961 to 2012. It covers $N_2O$ and $CH_4$ from various sources (e.g. rice cultivation, synthetic fertilizers, manure management). Because it covers a longer time period than other sources for the agricultural sector we use it with highest priority after the country reported data. The data is generated from activity data and emission factors following the Tier 1 approach of the IPCC 2006 guidelines.

Appendix B

20 gives details on the emissions categories and country processing.

**2.4 Region resolved datasets**

**2.4.1 Houghton land use $CO_2$ [HOUGHTON2008]**

This source covers land  cover change $CO_2$ emissions from 7 regions and three individual countries (USA, Canada, and China) for the years 1850 to 2005. The dataset is described in a series of papers by Houghton (2008, 2003, 1999). It is gener-

25 ated using a book keeping model to track carbon in living vegetation, dead plant material, wood products, and soils. The carbon stock and its changes are taken from field studies. Information on changes in land use are mostly taken from agricultural and forestry statistics, historical records, and national handbooks. Emissions outside tropical regions  since 1990 are estimates (constants[14]). For our dataset the regional emissions have to be downscaled to country level. This is described in Section 4.2.2, while technical details are given in Appendix B.
* * *
[14]The constant emissions outside tropical regions are obtained using the assumption that emissions calculated for 1990 are also valid for the subsequent years.

The dataset covers only direct (deliberate) human induced activities (Houghton (2003, 1999) ). Thus generally, forest fires are not included except for fire clearing. However, for the USA wildfires and the effect of measures for fire suppression are included.

We use this dataset as a proxy for $CO_2$ emissions from land use, land use change, and forestry (LULUCF) as land cover
5    change accounts for the majority of LULUCF emissions (Smith et al. (2014) ).

**2.4.2    RCP historical data [RCP]**

The Representative Concentration Pathways (RCPs) were created for the CMIP5 intercomparison study of Earth System Models for that was organized by the World Climate Research Programme and used (amongst others) in the IPCC's Fifth Assessment Report (AR5). They have a common historical emission time series, which covers all Kyoto gases but is only resolved
10    at a coarse regional and sectoral level (Meinshausen et al. (2011b)). For $N_2O$ and fluorinated gases only economy wide global emissions are available. For $CO_2$ global emissions are split into land use and fossil and industrial emissions. $CH_4$ emissions are resolved into 5 regions with several subcategories of the IPCC 1996 categorization.

RCP historical data are compiled from a wide range of emission sources and atmospheric concentration measurements. Where concentration data is used, inverse emission estimates are computed using the MAGICC6 reduced complexity climate
15    model (Meinshausen et al. (2011a) ). RCP historical data can be used for extrapolation of country time series to the past using regional growth rates. RCP land use emissions data is not used in our dataset as it is based on the Houghton dataset, which we include directly (see previous section). Preprocessing of RCP data is explained in Appendix B.

**2.5    EDGAR-HYDE 1.4 [EDGAR-HYDE14]**

The EDGAR-HYDE 1.4 "Adjusted Regional Historical Emissions 1890 – 1990" dataset covers the gases $CO_2$, $N_2O$, and $CH_4$
20    for the years 1890 to 1995 (Olivier and Berdowski (2001); Van Aardenne et al. (2001)). The data is given for 13 regions, some of which are individual countries (USA, Canada, Japan). It is generated from the EDGAR v3.2 dataset (Olivier and Berdowski (2001)) and the "Hundred Year Database for Integrated Environmental Assessments" (HYDE) (Van Aardenne et al. (2001); Goldewijk and Battjes (1997)). We use it to extrapolate country emissions into the past. It has a relatively high sectoral detail, but the sectors differ from the IPCC 1996 definitions, so mapping to IPCC 1996 sectors is necessary. Details are presented in
25    Appendix B.

**2.6    Gridded datasets**

The two gridded datasets included in the generation of the PRIMAPhist dataset do not contain any emissions data. Instead they contain data for potential vegetation and simulation data of past existing vegetation. By comparing these, we can determine areas where deforestation has occurred, which we use to downscale the Houghton land cover change emissions data to country
30    level. More information on the use of these datasets is provided in Section 4.2.2.

**2.6.1 HYDE land cover data [HYDE]**

[revised manuscript text omitted]

**Composite Source Generator**  The composite source generator (CSG) works on every country, gas, and category individually. Its core is the priority algorithm, which combines the sources following a given prioritization. The algorithm starts with the highest priority source. Missing time series are copied from lower priority sources. After this step the priority algorithm fills gaps in the time series using lower priority sources and extrapolates using year to year growth rates from lower priority sources. For each gas, category, country time series it is checked if the composite source contains gaps or does not cover the full time period. If that is the case the second highest priority source is checked for data that could fill gaps and extend the time series. If that time series itself contains gaps or needs extension, the hierarchy is parsed downwards recursively and the resulting time series is used to extend the composite source. For details on the harmonization see Appendix A4. For this study we add one source at a time and therefore do not parse the sources recursively but add what is present in the next source and then see if the resulting time series needs further extension. If there is data missing after the end of this process the CSG can numerically interpolate gaps and extrapolate missing data at the boundaries. For this dataset we only use the interpolation by the CSG, because we use regional growth rates from other sources to extrapolate the country data. A schematic of the composite source generator within the PRIMAP emissions module is shown in Figure 2. The rationale underlying this combination method is that the absolute values are taken from the highest priority source, while lower priority sources only used for the dynamics of emissions. By scaling the lower priority sources to match the higher priority source, we retain the year to year growth rates of the lower priority sources but adjust the absolute values to the highest priority source. For details see Appendix A4. Other options for harmonization are discussed in Section 8.

**Extrapolation**  Missing years in the past are extrapolated using growth rates from regional data or data from other sectors and numerical extrapolation. The details depend on gas and sector and are described later in this section. Missing data in the future is extrapolated linearly using a 15 years trend. This usually affects up to 4 years with very few exceptions where extrapolation is used for longer periods. We also offer a dataset without this numerical extrapolation.

When using extrapolation with growth rates from regions or other sectors we make the assumption that these time series share growth rates with the unknown time series we want to determine through extrapolation. This assumption seems crude, however it is much more transparent than e.g. building a more complicated model to compute the time series. A more sophisticated model will likely also need some input data like population or economic data to come up with extrapolated time series which is also scarce for the times we need the extrapolation for (i.e. before 1960 / 1970). Numerical extrapolation on the other hand does not use any information for the time we want to build our time series for, but only uses data from a range of years before or after the time period to be computed. It thus makes the assumption that we can deduce emissions in one time period from emission in another time period which is often not true. As an

[Figure]

**Figure 2.** The Composite  Source Generator (CSG) is used to assemble time series from different sources into one time series covering all countries, sectors, gases and years. The source prioritization in the figure is illustrative and does not represent the source prioritization for the dataset described here. In this study the internal category aggregation of the composite source generator is not used but categories are aggregated before the generation of the composite source to enable extrapolation of subcategories. For the PRIMAP-hist dataset we always combine only two sources at a time instead of recursively filling missing data. Section 4.1 and Appendix A desribe the use of the CSG for this dataset. figure 3 shows the individual steps for an example time series.

example consider the second world war, where emissions drastically changed, which a numerical extrapolation would not model when using e.g. 1960 - 1980 as input data. A regional time series for e.g. Europe, however, would have this feature and model emissions for european countries more realistical than numerical extrapolation. We still use numerical extrapolation for the PRIMAP-hist dataset, but only when it is the only option because no data exists.

**Postprocessing**  After extrapolation the individual gas and category time series are aggregated to build the higher categories and the Kyoto GHG basket. For details on the aggregation see Appendix A1.2.

Sudan needs a special treatment as the split into Sudan and South Sudan has been so recent that no separate emissions data is available yet. We downscale the Sudan emissions time series to Sudan and South Sudan using UN population statistics (UN Population Division (2015)) as a downscaling key. We also aggregate country data for some regional groups.

Figure 3 shows an example of how we build a pathway from different time series.

In the following we describe the data availability and use in detail for the different gases and sectors.

**CO$_2$**  Data coverage for CO$_2$ is in general very good. The largest emission sources are the consumption and production of fossil fuels and the production of cement. Both are covered by CDIAC, which extends the country reported data back to 1850 for 31 countries, back to 1900 for 65 countries, back to 1950 for 168 countries and to 1990 for 196 countries. For other sectors EDGAR42 extends the time series back to 1970. BP data completes the fossil fuel consumption time series until 2014.

To further extend time series into the past we use EDGAR-HYDE regional growth rates (starting in 1890). For categories 1A, 1B1, and 1B2 explicit time series are available while we use category 2 time series as a proxy for the subcategories of category 2. Other categories are not available. RCP CO$_2$ data that ranges back until 1850 is only available for total emissions excluding LULUCF on a global level. As total CO$_2$ emissions are dominated by fossil fuel burning we use the RCP data as growth rates to extrapolate category 1A emissions for those countries, which were not covered by CDIAC and EDGAR-HYDE from 1850 onwards. This does not affect any mayor emitter at the time for which data is extrapolated. For categories 3, 4, 6, and 7 no source for extrapolation is available so the first year is 1970 from EDGAR. We use growth rates of the the fossil fuel consumption time series for each country as a proxy to extend the time series of all other sectors to 1850.

The source prioritization and extrapolation is summarized in Table 2. Details of the growth rate extrapolation are discussed in Appendix A5.1.

**CH$_4$**  We have data on a per country level from 1990 to 2010 or 2012 from the country reported data. For agriculture (category 4) we have FAOSTAT data where the first year is 1961 and the last year 2012. For all other sectors and missing countries we use EDGAR42, which covers 1970 to 2010 for almost all countries. Categories 1, 2, 4, and 6 are extrapolated to 1890 using the regional growth rates from EDGAR-HYDE. The regional growth rates defined in the RCP historical database are used to extrapolate emissions in categories 1, 2, 4, and 6 back to 1850. Emissions in category 7 are extrapolated using

[Figure]

**Figure 3.** Example for the work of the composite source generator: the creation of the category 1A, $CO_2$ pathway for the Republic of Korea. The buildup starts with the UNFCCC source as there is no CRF data for the Republic of Korea. Extrapolation is not needed in this case, so the step is omitted from the figure. Details on the methodologies for the individual steps are given in Section 4.1 and Sections A4 and A5 of the appendix. The individual steps shown here correspond to the steps shown in Table 2.

| Step | Source | Categories | Countries | Years | Type of operation |
|------|--------|------------|-----------|-------|-------------------|
| 1 | CRF2014 | all | Annex I | 1990 - 2012 | CSG |
| 2 | CRF2013 | all | Annex I | 1990 - 2011 | CSG |
| 3 | UNFCCC2015 | all | 35 non-Annex I | 1990 - 2010 | CSG |
| 4 | BUR2015 | all | 8 non-Annex I | 1990 - 2012 | CSG |
| 5 | CDIAC2015 | 1A, 1B2, 2A | almost all | 1850 - 2011 | CSG |
| 6 | EDGAR42 | all | almost all | 1970 - 2010 | CSG |
| 7 | BP2015 | 1A | almost all | 1965 - 2014 | CSG |
| 8 | EDGAR-HYDE14 | 1A, 1B1-2, 2A-G | regions | 1890 - 1995 | growth rates extrap. |
| 9 | RCP | 1A | global | 1850 - 2005 | growth rates extrap. |
| 10 | PRIMAP-hist CAT1A | all but 1A | all | 1850 - 2014 | growth rates extrap. |
| 11 | numerical | all | all | 1850 - 2014 | linear extrapolation |

**Table 2.** Source prioritization and extrapolation for fossil and industrial $CO_2$. In Figure 3 we show the individual steps using the example of category 1 for the Republic of Korea. 
[revised manuscript text omitted]

starts as early as 1990. As the period from 1990 on is covered by FAOSTAT data this is no problem for us. In case countries still have missing data we extrapolate into the past using a linear pathway to zero emissions in 1850. Starting point of the extrapolation is a 21 year average. We use 0 in 1850 to rather under- than overestimate emissions when extrapolating. Linear or even exponential extrapolation is difficult for land use because of the strong fluctuations which strongly influence the trend that is needed for the extrapolation. This extrapolation is just used for very few small countries. 
[revised manuscript text omitted]

20 exchanges between countries. The CDIAC publication Andres et al. (1999) states that: "*Land exchanges between countries were also accommodated, when possible. For example, the emissions from Alsace-Lorraine were included with Germany or*

*France, reflecting which political unit governed these lands at any given time. This maintained the integrity of political entities despite changes in national borders."* This is not reflected in the country codes and thus remains in the final PRIMAP-hist dataset in contrast to the territorial accounting used in our methodology. We can not quantify the influence of this accounting discrepancy, because we do not know which regions were affected. However, as the land exchange including large emitters has been small in the recent decades and emissions have been relatively low before the recent decades the influence will likely be small. CRF2014, UNFCCC2015 and BUR2015 data are reported by countries and do not need  preprocessing as we use the territorial definitions of the UNFCCC reporting as a basis. For EDGAR data the rules regarding how emissions are assigned to countries in case of territorial changes are not clear from the methodology description and we assume that territorial accounting is used.

For some small countries and countries, which became recently independent no emissions data is available yet. In this case we have to construct it using other countries emissions data. Emission data for San Marino and the Vatican are included in Italian emissions data and downscaled using population shares.[16] This is done on the individual sources during preprocessing (see also Appendix B). For details see Appendix A3. Sudan and South Sudan are also downscaled from emissions of former Sudan using UN population data. (UN Population Division (2015)).

**5   Data availability**

The dataset is available from the GFZ Data Services under the url http://doi.org/10.5880/PIK.2016.003 (Gütschow et al. (2016)). When using this dataset or one of its updates, please cite this paper and the precise version of the dataset used. Please consider also citing the relevant original sources when using this dataset. Any use of this dataset should also comply with the usage restrictions of the original data sources used for this project.

**6   Results**

In this section we show some key results of our analysis. Details for additional countries, sectors and gases can be explored on-line on our companion website www.pik-potsdam.de/primap-live/primap-hist/. Here we focus on major emitters and global emissions.

**6.1   Sectoral distribution of aggregate Kyoto greenhouse gas emissions for major emitters**

Globally, production and consumption of fossil fuels is responsible for about two thirds of current  aggregate Kyoto greenhouse gas emissions (in the remainder of this section the term "emissions" refers to aggregate Kyoto GHG emissions). This is shown in the upper left panel of Figure 5 . An increase from about 50% in 1950 and a negligible contribution in 1850. Before the industrial revolution, deforestation was the major emissions source followed by agriculture. Currently, these sectors
* * *
[16]GDP data is not available.

are the second and third largest sources. Roughly 10% of emissions come from waste and industrial processes. Industrial processes increased their share in emissions after 1950 while the share of waste related emissions stayed relatively constant.

The sectoral profile differs strongly among countries (Figure 5). Land use emissions reached almost zero or even negative values in the 1950s to 1970s in industrialized countries (USA, EU, Japan) and a few decades later in China. For all these countries, fossil fuel use and production are the by far largest contributors to total emissions. While the industrialized countries show decreasing (USA, EU) or stagnating (Japan) fossil fuel emissions, China shows rapidly increasing emissions. The increase of emissions from china may have slowed down in the last years, but more time is needed to say if this is more than a temporary effect  (Korsbakken et al. (2016) ).

India still has a large share of LULUCF emissions with no clear increase or decrease in the last two decades. Agriculture and LULUCF have similar emissions both in trends and absolute values, which have only recently (roughly 1990) been surpassed by the steeply increasing fossil fuel related emissions. For Brazil the largest sector is land use, followed by agriculture. Land use emissions show a decreasing trend, but total emissions do not follow this trend due to a rise in agricultural emissions and fossil fuel related emissions.

Waste gives a small contribution, differing by country without a clear split between developed and developing countries. The contribution of industrial processes is larger in industrialized countries, but especially large in China.

**6.2    Gas distribution of economy wide emissions for major emitters**

The contribution of individual gases and gas groups to (GWP weighted) economy wide (IPCC 1996 category 0) emissions is shown in Figure 6. It is clearly visible that $CO_2$ is by far the largest contributor followed by $CH_4$ and $N_2O$, both globally and for individual countries. The contribution of fluorinated gases is in general small and negligible for developing countries. Again, China's emissions profile is closer to that of an industrialized country than to other major developing country emitters.  Economy wide methane emissions are high for countries with a large agricultural sector (India and Brazil). Japan is somewhat of an exception with almost all emissions from $CO_2$.

**7    Uncertainties**

In this paper we do not assess the uncertainties of the dataset in detail. Of the individual datasets used, uncertainty information is available for some while for others it is not provided. Where it is available, the level of detail is very different. Some datasets give per country or per regional group uncertainty estimates while others only provide global estimates. Individual uncertainty estimates can be over $100\%$ (Olivier et al. (1999) ). To calculate uncertainty estimates for all countries, gases and sectors for the composite source one has to transform the information given for the individual sources to a common methodology and level of detail and combine it in line with the creation of the composite source. As most datasets come without an uncertainty estimate and third party estimates are scarce for some datasets it is hard to find a consistent set of uncertainty estimates. Furthermore, different studies use different sectoral resolutions, confidence intervals etc., which makes it difficult to compare and combine the results to arrive at an estimate for our aggregate source. We leave this task for a future publication. In the following, we

[Figure]

**Figure 5.**  Aggregate Kyoto greenhouse gas emissions by sector for major emitters and the world. Where land use emissions are negative, the stacked emissions of the other sectors start at this negative value. International shipping and aviation emissions are not included. The figure is discussed in Section 6.1.

[Figure]

**Figure 6.**  Economy wide (IPCC 1996 category 0) emissions by gas for major emitters and the world. International shipping and aviation emissions are not included. The figure is discussed in Section **??**.

give a broad overview of the uncertainties of individual sources and

**7.0.1**

present an indicative uncertainty range for individual gases and sectors based on literature values. We plot this source together
5   with input data and the indicative uncertainty range to reveal differences between sources and identify possible problems (Figures 7, and 8).

**7.1 Uncertainties from individual sources**

Uncertainty estimates for the CDIAC dataset of global $CO_2$ emissions from fossil fuels and industry have varied since the first assessment made by Marland and Rotty (1984), which resulted in an uncertainty range between 6 and 10% (using a
10   90% confidence interval). In a recent publication a single global fossil fuel $CO_2$ emission uncertainty of 8.4% (using a 95% confidence interval) is offered as a reasonable combination of data (Andres et al. (2014)), in an attempt to simplify the different assessments and to make the best of the qualitative and quantitative knowledge developed since the first study of 1984.

Different approaches examine CDIAC global uncertainty as the aggregate of the uncertainties associated with fossil fuel $CO_2$ emissions from individual countries. In Andres et al. (1996)
15    a country grouping was introduced, which uses seven classes of countries with "*similar perceived uncertainty*". Andres et al. (2014) calculated uncertainty estimates for these groups, which are presented in Table 9.

The EDGAR team has stated that it was not feasible to go beyond the uncertainty tables compiled for EDGAR v2.0, where uncertainties are indicated in terms of ranges ranking from small (10%) to very large (> 100%) (PBL (2010)). However, other
20    institutions, such as UNEP (UNEP (2012)), estimated an uncertainty range of $\pm 10\%$ (for a 95% confidence interval) for total $CO_2$ (including LULUCF). For global emissions of $CH_4$, $N_2O$, and fluorinated gases uncertainties are estimated to be $\pm 25\%$, $\pm 30\%$, and $\pm 20\%$ respectively (using a 95% confidence interval) (UNEP (2012) ).

FAOSTAT  land use emission estimates are limited to only two carbon pools (above and below-ground biomass) out of six identified by the IPCC guidelines (above and below-ground, dead wood, litter, soil organic carbon, and harvested
25   wood products). Therefore, FAOSTAT estimates  greenhouse gas emissions and removals from land use are likely underestimated (Federici et al. (2015)).

Tubiello et al. (2015) provides overall uncertainty estimates of the FAOSTAT database, where global emission estimates from crop and livestock carry $\pm 30\%$ uncertainty ranges. Uncertainties in the land use sector are even larger, with a $\pm 50\%$ range.
30   Table 10 gives an overview of available uncertainty estimates for the individual sectors and gases included in the PRIMAP-hist dataset and how we calculated the indicative uncertainty range used in Figures 7 and 8 for different sectors and gases.

**7.1.1**

**7.2 Comparison with other data sources**

A different approach at uncertainty estimates is to compare different datasets. If they were completely independent, the distribution of emissions for the same category and gas should represent the uncertainties. This approach also captures uncertainties from different definitions of sectors, which are not included in the uncertainties of individual datasets. Here some sources are depending on each other or may use common underlying data, so we can not determine an upper bound on uncertainty, but rather a lower bound. Adding independent sources would likely increase uncertainty. In Figures 7 and 8 we  plot the composite source  alongside some of the individual sources and other composite sources for individual gases and sectors at a global level. To compare the lower bound of the inter-source uncertainty to the individual source uncertainty we also plot an indicative uncertainty range from Table 10 around the PRIMAP-hist dataset. It is apparent that for most categories and gases the inter-source uncertainty is lower than the uncertainty estimated for the individual sources.  However, as we have some source interdependence we cannot conclude that the individual uncertainties are overestimated. Additionally, the number of sources is too small to reliably sample the 95% confidence interval of the individual source uncertainty.

In the following we investigate discrepancies between sources for total emissions as well as individual gases and sectors to analyze if they are based on different assumptions and underlying data or lack of data for subsectors or individual gases. The EDGAR-HYDE data shows relatively low total Kyoto GHG values. The  sector plots show that this is due to low values for industrial processes and land use emissions. The low industrial process emissions can partly be explained by the lack of data for fluorinated gases in the EDGAR-HYDE dataset, but emissions for $CH_4$ and $CO_2$ are also low. Land use $CO_2$ emissions in the EDGAR-HYDE dataset are only about half of the emissions of all other datasets assessed and outside of the sizable uncertainty range applied to the PRIMAP-hist time series. We have to note that RCP, MATCH, and PRIMAP-hist include HOUGHTON data in their land use time series and are therefore not independent. The HOUGHTON based time series are consistent with EDGAR42 and FAO while the EDGAR-HYDE time series is not similar to any of the time series for more recent emissions.

A further major discrepancy is the RCP $CH_4$ time series, which differs strongly from all other sources. Emissions are significantly higher than in other sources but show a steep decline between 1990 and 2000. No other source used in this analysis shows this effect. RCP $CH_4$ emissions are based on Lamarque et al. (2010), which uses EDGAR-HYDE but adds information for some sectors missing in EDGAR-HYDE14, namely grassland and forest fire emissions.[17] However, the discrepancies can not fully be explained by this as they are present also in other sectors than land use.

For $N_2O$, MATCH and EDGAR42  aggregate Kyoto gas emissions are lower than the PRIMAP-hist dataset while EDGAR-HYDE14 and RCP are higher. MATCH uses EDGAR-HYDE (growth rates) prior to 1990, which explains the very similar pathway profiles and leads to very low emissions outside the uncertainty range before 1970.
* * *
[17]International shipping and aviation emissions are also added, but not included in this study.

[Figure]

**Figure 7.** Comparison of the PRIMAP-hist dataset with both individual sources and composite datasets for aggregate Kyoto gases and the main IPCC 1996 categories. Grey shaded areas show the indicative uncertainty range from Table 10 applied to the PRIMAP-hist dataset. Where different uncertainty estimates exist the average value is used. International shipping and aviation emissions are not included. The figure is discussed in Section 7.2.

[Figure]

**Figure 8.** Comparison of the PRIMAP-hist dataset with both individual sources and composite datasets for different gases. Grey shaded areas show the indicative uncertainty range from Table 10 applied to the PRIMAP-hist dataset. Where different uncertainty estimates exist the average value is used. International shipping and aviation emissions are not included. The figure is discussed in Section 7.2.

Finally, the estimates of emissions of fluorinated gases are higher for EDGAR42 than for our aggregate dataset. This means that country reported  fluorinated gas emissions are significantly lower than what EDGAR calculates the emissions to be.  Not all discrepancies between sources could be explained, and some are larger than the indicative uncertainty range for an individual source. This indicates that the actual uncertainties of emissions data could be even higher than what is assessed for individual sources.

**7.2.1**

**7.3 Uncertainties from methodology**

The creation of this composite dataset implies several decisions on source prioritization, extrapolations, and downscaling options. These questions usually do not have one "correct" solution but rather different options with individual benefits and drawbacks. Different options (e.g. linear or constant extrapolation) have different implications for the calculated emissions so the decisions introduce an "expert judgment uncertainty" to the final dataset. A further source of uncertainty is the use of regional growth rates for extrapolation. This assumes that all countries within that region shared the same growth rates, which is a simplification. Similarly downscaling uses simplifications like constant emission shares or the use of another source as a proxy. We only use these methods if no individual country data is available and have to accept the uncertainty to fill gaps in data. See also Section 8 below.

The scaling of one source to another also increases the uncertainties associated with the final time series compared to the individual time series: the uncertainty of the final time series can be calculated using standard error propagation formulas. For a scaling $f$ the standard deviation of a scaled time series $C = f \cdot B$ would be $s_c = \sqrt{s_f^2 + s_B^2}$, where $s_B$ is the standard deviation of the time series $B$ and $s_f$ is the standard deviation of the scaling factor, which depends on $s_B$ and $s_A$ in a manner determined by the exact matching algorithm ($A$ denotes the time series, which $B$ is adjusted to).

[revised manuscript text omitted]

Le Quéré, C., Moriarty, R., Andrew, R. M., Canadell, J. G., Sitch, S., Korsbakken, J. I., Friedlingstein, P., Peters, G. P., Andres, R. J., Boden, T. A., Houghton, R. A., House, J. I., Keeling, R. F., Tans, P., Arneth, A., Bakker, D. C. E., Barbero, L., Bopp, L., Chang, J., Chevallier, F., Chini, L. P., Ciais, P., Fader, M., Feely, R. A., Gkritzalis, T., Harris, I., Hauck, J., Ilyina, T., Jain, A. K., Kato, E., Kitidis, V., Klein Goldewijk, K., Koven, C., Landschützer, P., Lauvset, S. K., Lefèvre, N., Lenton, A., Lima, I. D., Metzl, N., Millero, F., Munro, D. R., Murata, A., Nabel, J. E. M. S., Nakaoka, S., Nojiri, Y., O'Brien, K., Olsen, A., Ono, T., Pérez, F. F., Pfeil, B., Pierrot, D., Poulter, B.,

Rehder, G., Rödenbeck, C., Saito, S., Schuster, U., Schwinger, J., Séférian, R., Steinhoff, T., Stocker, B. D., Sutton, A. J., Takahashi, T., Tilbrook, B., van der Laan-Luijkx, I. T., van der Werf, G. R., van Heuven, S., Vandemark, D., Viovy, N., Wiltshire, A., Zaehle, S., and Zeng, N.: Global Carbon Budget 2015, Earth System Science Data, 7, 349–396, doi:10.5194/essd-7-349-2015, http://www.earth-syst-sci-data.net/7/349/2015/, 2015.

5   Levin, I., Naegler, T., Heinz, R., Osusko, D., Cuevas, E., Engel, A., Ilmberger, J., Langenfelds, R. L., Neininger, B., Rohden, C. V., Steele, L. P., Weller, R., Worthy, D. E., and Zimov, S. a.: The global SF6 source inferred from long-term high precision atmospheric measurements and its comparison with emission inventories, Atmospheric Chemistry and Physics, 10, 2655–2662, doi:10.5194/acp-10-2655-2010, 2010.

Liu, Y. Y., van Dijk, A. I. J. M., de Jeu, R. a. M., Canadell, J. G., McCabe, M. F., Evans, J. P., and Wang, G.: Recent reversal in loss of global terrestrial biomass, Nature Climate Change, 5, 1–5, doi:10.1038/nclimate2581, http://www.nature.com/doifinder/10.1038/nclimate2581%5Cnpapers3://publication/doi/10.1038/nclimate2581, 2015.

Luo, G. J., Kiese, R., Wolf, B., and Butterbach-Bahl, K.: Effects of soil temperature and moisture on methane uptake and nitrous oxide emissions across three different ecosystem types, Biogeosciences, 10, 3205–3219, doi:10.5194/bg-10-3205-2013, 2013.

Marland, G. and Rotty, R. M.: Carbon dioxide emissions from fossil fuels: a procedure for estimation and results for 1950-1982, Tellus B, 36B, 232–261, doi:10.1111/j.1600-0889.1984.tb00245.x, 1984.

15   Matthews, H. D., Graham, T. L., Keverian, S., Lamontagne, C., Seto, D., and Smith, T. J.: National contributions to observed global warming, Environmental Research Letters, 9, 014 010, doi:10.1088/1748-9326/9/1/014010, http://stacks.iop.org/1748-9326/9/i=1/a=014010?key=crossref.4f98ca366f68bd27560fb48645330938, 2014.

Meinshausen, M.: RCP Concentration Calculations and Data Final Version, background data, acknowledgements and further info, http://www.pik-potsdam.de/~mmalte/rcps/index.htm, 2011.

20   Meinshausen, M. and Alexander, R.: INDC Factsheets, climatecollege.unimelb.edu.au/indc-factsheets, 2016.

Meinshausen, M., Raper, S. C. B., and Wigley, T. M. L.: Emulating coupled atmosphere-ocean and carbon cycle models with a simpler model, MAGICC6 – Part 1: Model description and calibration, Atmospheric Chemistry and Physics, 11, 1417–1456, doi:10.5194/acp-11-1417-2011, http://www.atmos-chem-phys.net/11/1417/2011/, 2011a.

Meinshausen, M., Smith, S. J., Calvin, K., Daniel, J. S., Kainuma, M. L. T., Lamarque, J.-F., Matsumoto, K., Montzka, S. a., Raper,
25   S. C. B., Riahi, K., Thomson, A., Velders, G. J. M., and Vuuren, D. P.: The RCP greenhouse gas concentrations and their extensions from 1765 to 2300, Climatic Change, 109, 213–241, doi:10.1007/s10584-011-0156-z, http://www.springerlink.com/index/10.1007/s10584-011-0156-z, 2011b.

Meinshausen, M., Jeffery, L., Guetschow, J., Robiou du Pont, Y., Rogelj, J., Schaeffer, M., Höhne, N., den Elzen, M., Oberthür, S., and Meinshausen, N.: National post-2020 greenhouse gas targets and diversity-aware leadership, Nature Climate Change, pp. 1–10,
30   doi:10.1038/nclimate2826, http://www.nature.com/doifinder/10.1038/nclimate2826, 2015.

[revised manuscript text omitted]

UNEP: The Emissions Gap Report 2012, Tech. rep., United Nations Environment Programme, http://www.unep.org/pdf/2012gapreport.pdf, 2012.

UNEP: The Emissions Gap Report 2014, UNEP, http://www.unep.org/emissionsgapreport2014/, 2014.

UNEP: The Emissions Gap Report 2015, United Nations Environment Programme, http://www.unep.org/emissionsgapreport2015/, 2015.

UNFCCC: National Inventory Submissions 2013, http://unfccc.int/national_reports/annex_i_ghg_inventories/national_inventories_submissions/items/7383.php, 2013.

UNFCCC: National Inventory Submissions 2014, http://unfccc.int/national_reports/annex_i_ghg_inventories/national_inventories_submissions/items/8108.php, 2014a.

UNFCCC: 6th National Communications to the UNFCCC by AnnexI parties, http://unfccc.int/national_reports/annex_i_natcom/submitted_natcom/items/7742.php, 2014b.

UNFCCC: National Communications to the UNFCCC by non-AnnexI parties, http://unfccc.int/national_reports/non-annex_i_natcom/submitted_natcom/items/653.php, 2015a.

UNFCCC: UNFCCC Decision 1/CP.21: Paris Agreement, Tech. rep., United Nations Framework Convention on Climate Change, http://unfccc.int/resource/docs/2015/cop21/eng/10a01.pdf, 2015b.

UNFCCC: UNFCCC Detailed by Party data, http://unfccc.int/di/DetailedByParty.do, 2015c.

UNFCCC: Submitted biennial update reports (BURs) from non-Annex I Parties, http://unfccc.int/national_reports/non-annex_i_natcom/reporting_on_climate_change/items/8722.php, 2016.

United Nations: 2013 Energy Statistics Yearbook, United Nations Department for Economic and Social Information and Policy Analysis, Statistics Division, New York, http://unstats.un.org/unsd/energy/yearbook/, 2016.

[revised manuscript text omitted]
 1% with the exception of category 1 where rounding is to 0.1%. For category 5 the high and low uncertainty cases are the same within rounding, so only one number is given.

| EDGAR-HYDE | IPCC1996 |
|---|---|
| FNN | CAT1A |
| FPP | CAT1B |
| I00 | CAT2 |
| LGG + LNN + L42 + L43 + L70 | CAT4 |
| L41 | CAT5 |
| W10 + WNN | CAT6 |

**Table 11.** Category matching for EDGAR-HYDE and IPCC 1996 categories.